# Hyperbolic shear polaritons in low-symmetry crystals

Nikolai C. Passler[1], Xiang Ni[2], Guangwei Hu[2,4], Joseph R. Matson[5], Giulia Carini[1], Martin Wolf[1], Mathias Schubert[6], Andrea Alù[2,3 ✉], Joshua D. Caldwell[5 ✉], Thomas G. Folland[7 ✉] & Alexander Paarmann[1 ✉]

The lattice symmetry of a crystal is one of the most important factors in determining its physical properties. Particularly, low-symmetry crystals offer powerful opportunities to control light propagation, polarization and phase[1–4]. Materials featuring extreme optical anisotropy can support a hyperbolic response, enabling coupled light–matter interactions, also known as polaritons, with highly directional propagation and compression of light to deeply sub-wavelength scales[5]. Here we show that monoclinic crystals can support hyperbolic shear polaritons, a new polariton class arising in the mid-infrared to far-infrared due to shear phenomena in the dielectric response. This feature emerges in materials in which the dielectric tensor cannot be diagonalized, that is, in low-symmetry monoclinic and triclinic crystals in which several oscillators with non-orthogonal relative orientations contribute to the optical response[6,7]. Hyperbolic shear polaritons complement previous observations of hyperbolic phonon polaritons in orthorhombic[1,3,4] and hexagonal[8,9] crystal systems, unveiling new features, such as the continuous evolution of their propagation direction with frequency, tilted wavefronts and asymmetric responses. The interplay between diagonal loss and off-diagonal shear phenomena in the dielectric response of these materials has implications for new forms of non-Hermitian and topological photonic states. We anticipate that our results will motivate new directions for polariton physics in low-symmetry materials, which include geological minerals[10], many common oxides[11] and organic crystals[12], greatly expanding the material base and extending design opportunities for compact photonic devices.

Crystal symmetry plays a critical role in dictating the optical, electronic, mechanical and thermal properties of a material. Reduced symmetry is at the heart of numerous emergent phenomena, including structural phase transitions[11], charge-density waves[13] and topological physics[14]. The interaction of light with low-symmetry materials is particularly important, as it allows fine control over the phase, propagation direction and polarization[1–4]. This control can be especially pronounced for sub-diffractional surface waves, for instance, surface phonon polaritons (SPhPs)[15] and surface plasmon polaritons (SPPs), supported at the surface of polar crystals and conductors, respectively. Both SPhPs and SPPs are quasiparticles comprising photons and coherently oscillating charges, that is, polar lattice vibrations or free-carrier plasmas, respectively, and they are strongly influenced by crystal symmetry. As a relevant example, low-symmetry polaritonic materials can support hyperbolic light propagation[16], constituting an exotic class of light waves that are highly directional with very large momenta. Hyperbolic polaritons occur in materials in which the real part of the permittivity along at least one crystal direction is negative

and positive along at least one other. This extreme anisotropy is associated with free carriers and optic phonons in anisotropic lattices. In turn, hyperbolic polaritons enable deeply sub-wavelength light confinement over broad bandwidths[8,9]. In polar crystals with symmetries that support a single optical axis (uniaxial), such as hexagonal boron nitride (hBN), hyperbolic polaritons (HPs) of type I or type II can arise[5,8,9], for which the hyperbolic isofrequency surfaces do or do not intersect the optical axis, respectively. Materials or metamaterials exhibiting lower symmetry, in which all three major polarizability axes are different (biaxial) but orthogonal, such as alpha-phase molybdenum trioxide ($\alpha$-MoO$_3$)[1,3], Li-intercalated vanadium pentoxide (V$_2$O$_5$)[4] or nanostructured metasurfaces[17], exhibit several distinct spectral regimes of hyperbolic modes propagating along different crystal axes. Notably, in-plane hyperbolicity within $\alpha$-MoO$_3$ films has been shown to be low loss[1,3], with reconfigurable features[2] and capable of supporting topological transitions[2]. More exotic polaritonic responses may be expected in crystals with further reduced symmetry, such as monoclinic and triclinic lattices.

[1]Fritz Haber Institute of the Max Planck Society, Berlin, Germany. [2]Photonics Initiative, Advanced Science Research Center, City University of New York, New York, NY, USA. [3]Physics Program, Graduate Center, City University of New York, New York, NY, USA. [4]Department of Electrical and Computer Engineering, National University of Singapore, Singapore, Singapore. [5]Vanderbilt University, Nashville, TN, USA. [6]University of Nebraska, Lincoln, NE, USA. [7]The University of Iowa, Iowa City, IA, USA. ✉e-mail: aalu@gc.cuny.edu; josh.caldwell@vanderbilt.edu; thomas-folland@uiowa.edu; alexander.paarmann@fhi-berlin.mpg.de

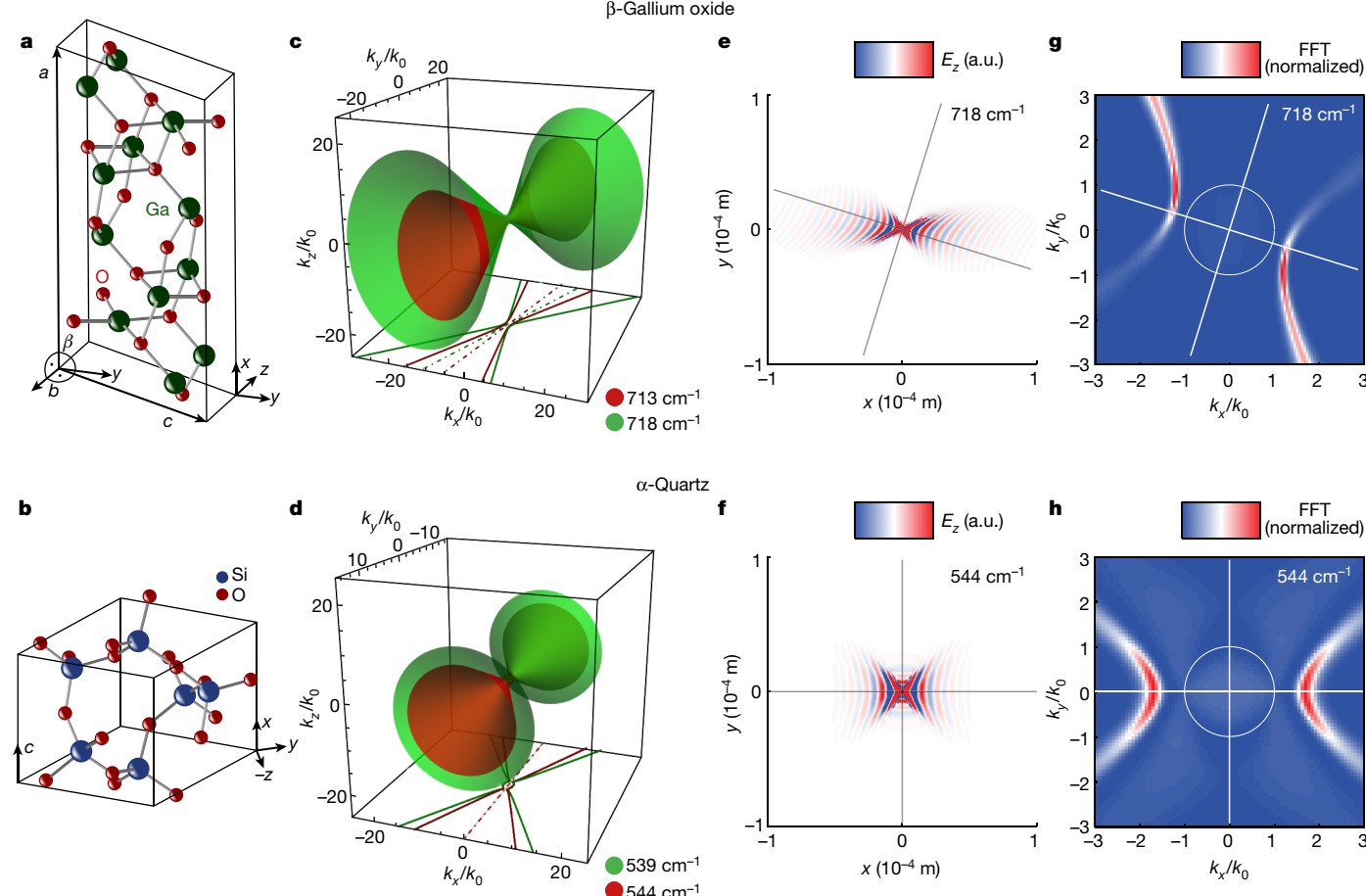

**Fig. 1 | Hyperbolic shear polaritons in monoclinic bGO compared with hyperbolic polaritons in uniaxial aQ. a**, Monoclinic crystal structure of bGO (monoclinic angle $\beta = 103.7°$). The sample surface of the investigated bGO crystal is the monoclinic (010) plane (x–y plane). **b**, Rhombic unit cell of aQ with the c axis oriented along the x direction, lying in the sample surface plane. The Cartesian coordinate system used in this work is shown. **c**, **d**, Isofrequency surfaces for bGO and aQ, respectively, at two frequencies (red and green) in which the material supports hyperbolic polaritons. The contour lines at $k_z = 0$ are plotted as solid lines at the bottom and their mirror axes are shown as dash-dotted lines. **e**, **f** Real-space electric fields at the bGO and aQ surfaces, respectively. **g**, **h** The respective two-dimensional Fourier transformation. Lines indicate the frequency-dependent optical axes for bGO and the crystal axes for aQ. FFT, fast Fourier transform.

Monoclinic crystals make up the largest crystal system, with around one-third of the minerals of Earth belonging to one of its three classes[18]. These low-symmetry Bravais lattices exhibit non-orthogonal principal crystal axes (Fig. 1a), in contrast to orthorhombic (for example, biaxial α-MoO₃ (ref. [1])), tetragonal, hexagonal, trigonal (for example, uniaxial α-quartz, aQ, Fig. 1b) or cubic crystal systems. As a consequence, their dielectric permittivity tensor has major polarizability directions that strongly depend on the frequency, with off-diagonal terms that cannot be completely removed through coordinate rotation[6,7], and exhibits shear terms analogous to viscous flow[19]. These features arise due to the non-trivial relative orientation (neither parallel nor orthogonal) of several optical transitions that, at a given frequency, contribute to a net polarization that cannot be aligned with the crystal axes. In turn, this property results in exotic light propagation not supported by higher-symmetry crystals[6,7,20]. Here we show exemplary consequences of these material features for nanophotonics, in particular, the emergence of a new form of waves − hyperbolic shear polaritons (HShPs) − which have not been previously observed.

In this work, we theoretically and experimentally demonstrate the emergence of HShPs in monoclinic crystals. As an exemplary material to demonstrate this phenomenon, we study beta-phase Ga₂O₃ (bGO), which has gained a large amount of research and industrial attention for its high breakdown field[21] and applications in photovoltaics[22], optical

displays[23] and gas sensors[24]. In the low-energy range, bGO features several strong infrared-active, non-orthogonal phonon resonances[6], making the permittivity tensor of bGO naturally non-diagonalizable. Its low symmetry has two consequences on the polariton propagation when compared with more conventional hyperbolic materials with a diagonal permittivity tensor, such as hBN, aQ and α-MoO₃. First, both the bGO polariton wavelength and the propagation direction strongly disperse with frequency. Second, we demonstrate that the asymmetric nature of optical loss in such crystals gives rise to shear, resulting in polariton propagation with tilted wavefronts. Such tilted wavefronts are a direct consequence of the low symmetry of the material and are one of the most notable and unique features of HShPs. New opportunities for polaritonics arise for HShPs stemming directly from their non-Hermitian and topological nature. Yet, surprisingly, they can be observed in low-loss, naturally occurring materials, without the need for artificial structuring of a material surface[17].

To highlight the role of the asymmetry of monoclinic crystals in their polariton response, we compare HShPs with HPs supported by higher-symmetry anisotropic crystals, such as aQ[25]. In this vein, we compare the crystal structure of monoclinic bGO in Fig 1a with the trigonal crystal of aQ in Fig. 1b, illustrating the low crystal symmetry present in bGO. In general, the description of the dielectric response of monoclinic crystals requires inclusion of identical off-diagonal

elements in the monoclinic plane within the frequency-dependent, complex-valued dielectric tensor.

$$\overline{\overline{\varepsilon(\omega)}} = \begin{bmatrix} \varepsilon_{xx}(\omega) & \varepsilon_{xy}(\omega) & 0 \\ \varepsilon_{xy}(\omega) & \varepsilon_{yy}(\omega) & 0 \\ 0 & 0 & \varepsilon_{zz}(\omega) \end{bmatrix} \qquad (1)$$

The coordinate systems used to define the response of bGO and aQ are sketched in Fig. 1a, b, respectively. To analyse the properties of HShPs in monoclinic materials, we first rigorously solve Maxwell's equations (see Methods) to calculate the dispersion relation of the polaritonic modes supported by bGO and — for comparison — aQ, each at two distinct frequencies. Initially, we consider the lossless case, in which the imaginary part of each term in the dielectric tensor is neglected for both bGO and aQ. The solutions for the polariton wavevectors in both materials at two different frequencies are provided in Fig. 1c, d. For aQ, we observe two open hyperboloid surfaces — as expected for uniaxial hyperbolic materials — in which a change in frequency results in a corresponding change in wavevector, while preserving the hyperboloid orientation, that is, the direction of polariton propagation (Fig. 1d). By contrast, as we change the frequency, not only does the bGO polariton wavevector magnitude change but the direction of the hyperboloid also rotates within the monoclinic plane, as can be appreciated by examining the $k_z = 0$ projections (Fig. 1c). This is a direct consequence of the non-trivial relative orientation of the phonon resonances supporting the hyperbolic response[6], which results in polariton bands that disperse in azimuth angle as a function of frequency. This feature represents a signature of the reduced symmetry associated with HShPs supported in monoclinic crystals (and is also anticipated in triclinic crystals), in contrast to HPs observed in higher-symmetry lattices.

When we also account for natural material losses resulting from inherent phonon-scattering processes, the polariton propagation in bGO shows a reduced symmetry in comparison with hyperbolic polaritons in aQ, even at individual frequencies, as illustrated in Fig. 1e, f. In these panels, we show the results of full-wave calculations of dipole-launched surface polaritons propagating across the surface of a semi-infinite slab of bGO and y-cut aQ, in which — in both cases — natural material losses were explicitly taken into account. For in-plane hyperbolic materials, these surface waves show a hyperbolic dispersion within the surface plane and are referred to as hyperbolic surface or hyperbolic Dyakonov polaritons[26,27], constituting a subset of HPs supported in these materials similar to volume-confined HPs in thin films. For aQ, HPs spread out along one crystal axis of the surface and are symmetric with respect to the crystal axes, as can be confirmed by a Fourier transform of the real-space electric field profile (Fig. 1h). However, for bGO (Fig. 1e), we observe that HPs are rotated with respect to the coordinate system of the monoclinic plane, as anticipated by the isofrequency contours (Fig. 1c). In addition, the wavefronts are tilted with respect to the major propagation direction, with no apparent mirror symmetry. This feature can also be clearly seen by examining the Fourier transform of the real-space profile (Fig. 1g), exhibiting a stronger intensity along one side of the hyperbola. These observations constitute the discovery of HShPs in low-symmetry crystals.

To experimentally demonstrate the effects of reduced symmetry in polariton propagation in bGO in contrast to higher-symmetry materials, we compare the azimuthal dispersion of HShPs in bGO to the one of HPs in aQ using an Otto-type prism-coupling geometry[28,29] (sketched in Fig. 2a; for details, see Methods). The experimental azimuthal dispersion of HPs on the surface of aQ is shown in Fig. 2b (see also Extended Data Fig. 3), in excellent agreement with the corresponding simulations (Fig. 2c). The dips in the reflectance spectra show the polariton resonances, which are only observable along specific azimuth angles and are symmetric about the crystal axes, $\phi = 0°$ (180°) and 90°. By contrast, the experimental azimuthal dispersion of HShPs on monoclinic bGO

(Fig. 2d) exhibits no mirror symmetry, again in excellent agreement with the simulated dispersion (Fig. 2e).

To experimentally access the in-plane hyperbolic dispersion of the HShP in bGO observed in Fig. 1e, we mapped out the frequency–momentum dispersion in close spectral proximity of that mode (680–720 cm⁻¹) at many azimuth angles (see Extended Data Fig. 4). The resulting map of polariton resonance frequencies is shown in Fig. 2f, in excellent agreement with the simulated resonance frequencies shown in Fig. 2g. These data allow extraction of single-frequency in-plane dispersion curves shown in Fig. 2h, i from experiment and simulations, respectively, for several selected frequencies, clearly demonstrating a hyperbolic dispersion, in excellent agreement with Fig. 1e. Notably, the base of the hyperbola shifts continuously with frequency, as marked by the symmetry axes for each curve in Fig. 2i, which directly leads to an asymmetric distribution of the group velocity along the hyperbolic dispersion curve, as shown in Fig. 2j (see also Extended Data Fig. 7).

The reduced symmetry observed in the polaritonic dispersion of bGO (Fig. 2d) is a direct consequence of the lack of symmetry in its vibrational structure[6]. Therefore, the HShPs are not propagating along fixed axes but show a continuous rotation of the HShP propagation direction as the frequency is varied. To describe the nature of this rotation, we diagonalize the real part of the permittivity tensor of bGO $\mathrm{Re}[\overline{\overline{\varepsilon(\omega)}}]$ individually at each frequency, by rotating the monoclinic plane by the frequency-dependent angle.

$$\gamma(\omega) = \frac{1}{2}\arctan\left(\frac{2\mathrm{Re}(\varepsilon_{xy})(\omega)}{\mathrm{Re}(\varepsilon_{xx})(\omega) - \mathrm{Re}(\varepsilon_{yy})(\omega)}\right) \qquad (2)$$

The dispersion of $\gamma(\omega)$ is shown in Fig. 2e (white lines), illustrating that the major polarizability directions within the monoclinic plane, denoted as m and n, vary widely across the range. This frequency-dependent coordinate system enables an easier understanding and classification of the polaritonic response (see Methods and Extended Data Fig. 1 for details). The rotated coordinate axes are shown in Fig. 1e, g (see also Extended Data Fig. 2 for further modes), illustrating their alignment with the hyperbolic dispersion.

Although equation (2) describes the frequency variation of the polariton propagation direction, it does not capture the tilted wavefronts observed in Fig. 1e. This is because, as we choose the rotated coordinate system [mnz], we still retain a purely imaginary off-diagonal permittivity component (see Extended Data Fig. 1). These terms are associated with the non-orthogonal relative orientation of the material resonances, coupling the two crystal axes in the monoclinic plane. As a result, even in the rotated coordinate system [mnz], the dielectric tensor has off-diagonal terms associated with shear phenomena.

To selectively analyse the role of these shear terms, we simulate the polariton propagation in the rotated coordinate system [mnz] at 718 cm⁻¹. In particular, we include a scaling factor for the magnitude of the off-diagonal imaginary component, indicated as $i \times f\mathrm{Im}(\varepsilon_{mn})$, with $f = 0$, 0.5 and 1 (shown in Fig. 3a–c), while retaining the diagonal loss terms. When we remove the off-diagonal component ($f = 0$), bGO essentially becomes a shear-free biaxial material, akin to α-MoO₃ and similar to uniaxial aQ, with polaritons propagating along the optical axes (Fig. 3a). Therefore, whereas polariton propagation in such a fictional form of bGO is anisotropic in specific spectral ranges (similar to polaritons in MoO₃ (refs. [1,3])), mode propagation without shear phenomena is symmetric about the (frequency-dependent) major polarizability axes (Fig. 3a). As we gradually increase the magnitude of the off-diagonal imaginary terms back to its natural value ($f = 1$), the wavefronts become increasingly skewed from the major polarizability axis (Fig. 3b, c). The respective reciprocal space maps in Fig. 3d–f show a strong symmetry breaking in the intensity distribution within the hyperbolic isofrequency curves. This observation provides further evidence that the propagation of polaritons is non-trivial within low-symmetry monoclinic — and, by extension, triclinic — systems, and it cannot be

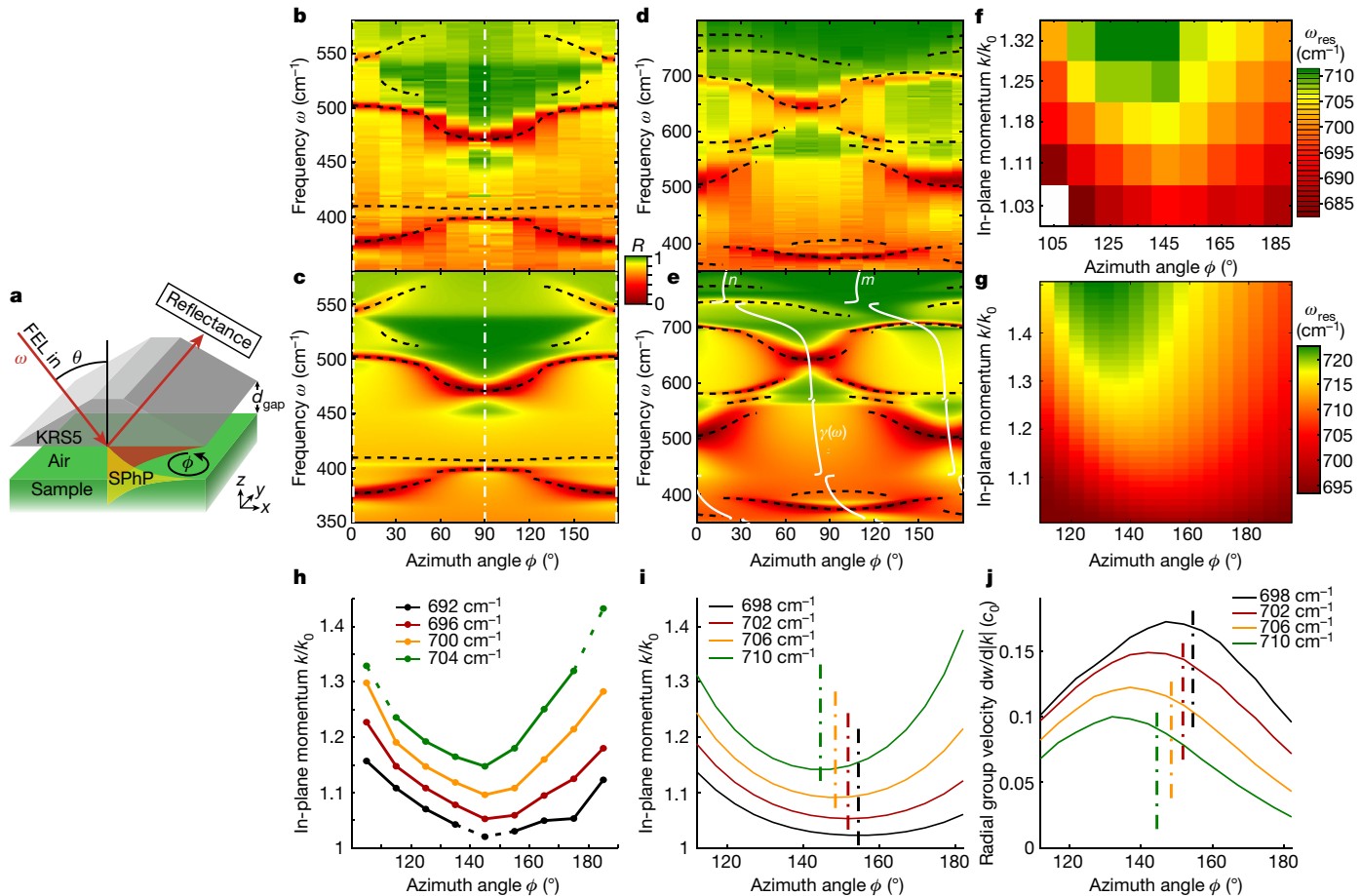

**Fig. 2 | Experimental observation of HPs on aQ and HShPs on bGO.**
**a**, Otto-type prism-coupling configuration for the experimental observation of surface waves. The p-polarized free-electron laser (FEL) excitation beam is reflected at the prism back side at an incident angle of $\theta = 28°$. The reflectance detection is unpolarized. The gap size was fixed to $d_{gap} \approx 10.4\,\mu m$ for aQ and $d_{gap} \approx 8.3\,\mu m$ for bGO. Datasets at other gap sizes are shown in Extended Data Fig. 3. **b**, Experimental azimuth dependence of HPs on aQ. **c**, Corresponding simulated reflectance map calculated by means of a transfer matrix method. **d**, Experimental azimuth dependence of HShPs on bGO. **e**, Corresponding simulated reflectance map. The white lines in **e** correspond to the directions of the frequency-dependent major polarizability axes $n$ and $m$; see text.

**f**, Experimental polariton resonance frequency map for bGO in the 680–720 cm$^{-1}$ frequency range, extracted from Otto reflectance measurements at various incidence angles $\theta$ and azimuth angles $\phi$. Experiments were performed at constant gap size $d_{gap} \approx 4.0\,\mu m$. **g**, Simulated polariton resonance frequency map. Experimental (**h**) and simulated (**i**) in-plane hyperbolic dispersion for bGO at selected frequencies interpolated from **f** and **g**, respectively. Dashed lines mark extrapolated values outside the accessed momentum range. **j**, Radial component of the group velocity extracted from **g**. Dash-dotted lines in **i** and **j** mark the symmetry axis for each dispersion curve, which shift with frequency. The asymmetric distribution of the radial group velocity shows the asymmetry of energy flow for HShPs.

expected in higher-symmetry materials in which polariton propagation patterns are symmetric about the principal crystal axes[1,3,4].

To connect the reduced symmetry of the surface subset of HShPs observed here experimentally (Fig. 2) and through our simulations (Fig. 3a–c) to the more general HShPs in the bulk, we now calculate isofrequency surfaces for polariton modes in bGO explicitly including loss, to account for the effect of shear phenomena. To this end, we solve Maxwell's equations for real momentum values, yielding complex frequency eigenvalues, whose imaginary part accounts for the finite lifetime of the supported modes (see Methods for details). The results of these calculations are shown in Fig. 3g–i. Here the real part of the eigenfrequency is fixed and we find its imaginary part $\omega_i$ (Fig. 3h), which is proportional to the modal lifetime, and the corresponding value of $k_z$ (Fig. 3g) for each pair of $k_m$ and $k_n$. The calculations are performed in the rotated coordinate system for both $f = 0$ and $f = 1$, showing that both the shape of the isofrequency surfaces as well as their lifetimes change greatly with the inclusion of the off-diagonal imaginary components. Notably, these calculations prove that, at individual frequencies and in the major polarizability frame, mirror symmetry of polariton propagation is lost in monoclinic materials as a direct consequence of shear.

To relate the isofrequency contours of HShPs to the surface mode dispersions in Fig. 3d–f, we plot the $k_z = 0$ solution in Fig. 3i, with the colour scale indicating the relative loss $\omega_i$ of the mode. Two important observations can be made: first, the mirror symmetry of the isofrequency curves is broken for $f > 0$ and it requires higher-order terms to account for the asymmetric shape; second, the mode losses are redistributed asymmetrically, with losses decreasing in one arm of the hyperbolae but increasing on the other arm. We note that, also in the experimental data (Extended Data Fig. 4), we observe an indication of asymmetric distribution of polariton quality factors along the hyperbolic dispersion curves (see Extended Data Fig. 5).

These observations naturally link HShPs in monoclinic crystals to the rich, emerging area of non-Hermitian and topological photonics. Although loss in orthogonal systems alone can already have interesting consequences for polariton propagation[30], the off-diagonal shear terms highlighted here can provide new opportunities for non-Hermitian photonics and for manipulation of topological polaritons in low-symmetry materials. For instance, we foresee asymmetric topological transitions experienced by HShPs, generalizing previous results in orthorhombic systems[2] by exploiting the unique

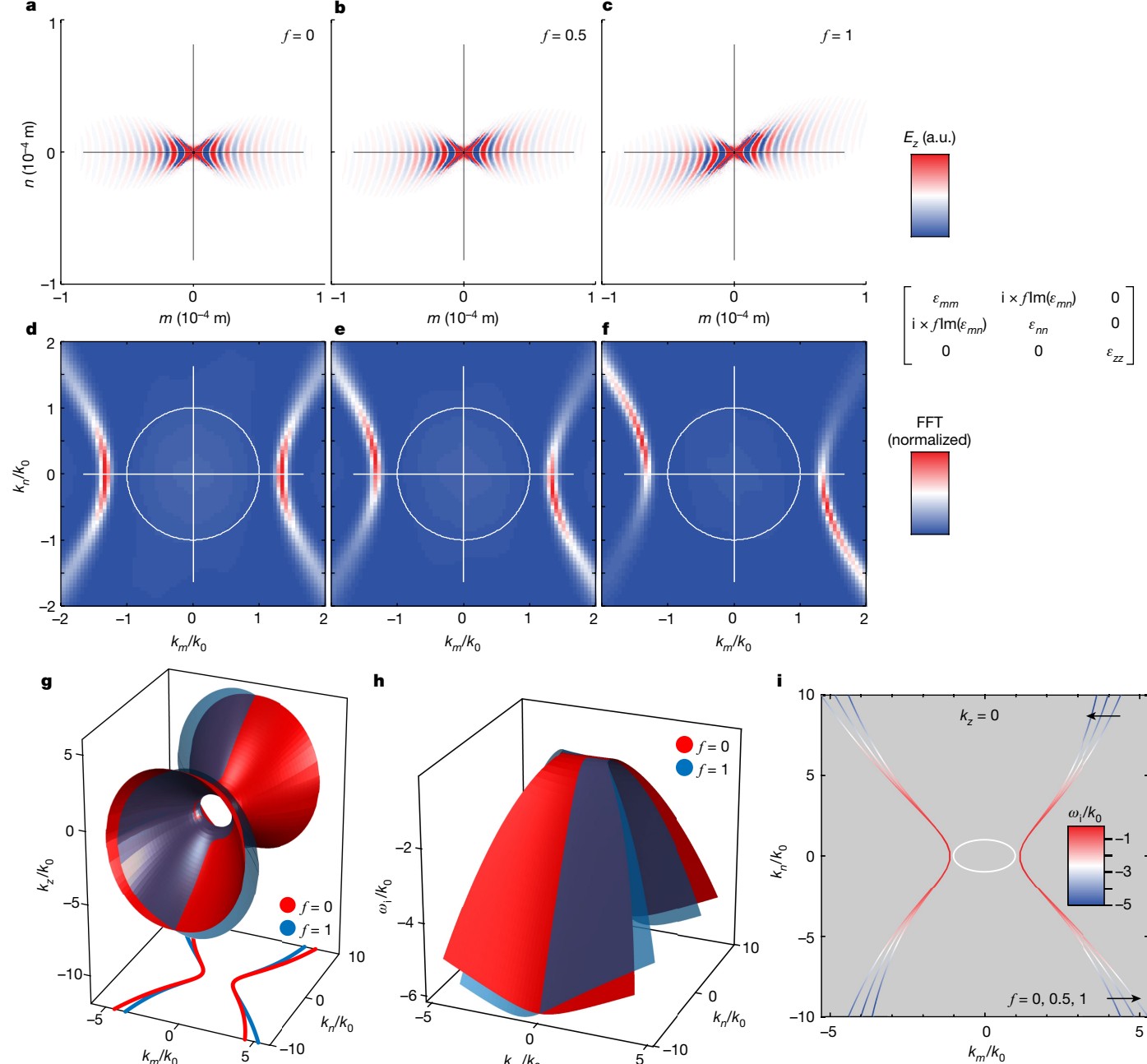

**Fig. 3 | Symmetry breaking by shear phenomena for hyperbolic polaritons in monoclinic bGO. a–c,** Real-space electric fields at the bGO surface for three different magnitudes of the off-diagonal permittivity element i × $f$Im($\varepsilon_{mn}$) calculated in the frequency-dispersive rotated coordinate system at 718 cm$^{-1}$. **d–f,** Respective Fourier transformation $k$-space maps in arbitrary units. The thin crosshairs indicate the principal axes in the frequency-dispersive rotated coordinate system, aligned horizontally and vertically because the maps were calculated within the rotated coordinate system [$mnz$]. **g,** Isofrequency surface of bulk eigenmodes in bGO with complex frequency $\tilde{\omega} = \omega/k_0 = 1 + i\omega_i/k_0$ calculated in the rotated frame for $f = 0$ and 1 (red and blue, respectively). See Methods for details on the approach. **h,** Imaginary part $\omega_i$ for $f = 0$ and 1 (red and blue, respectively). **i,** Contour lines of the isofrequency surface at $k_z = 0$ for $f = 0$, 0.5 and 1. The imaginary part $\omega_i$ at the corresponding point in $k$-space is colour-coded. FFT, fast Fourier transform.

non-Hermitian features emerging in low-symmetry materials. In addition, recent studies suggest the connection between Dyakonov surface waves and surface states emerging from one-dimensional band degeneracy (nodal lines) of topological nature of high-symmetry metacrystals[31]. We anticipate that HShPs may generalize these opportunities to asymmetric topological bands in which non-Hermiticity in the natural materials plays a dominant role.

Here we have demonstrated that low-symmetry crystals can support a new class of hyperbolic polariton modes with broken symmetry due to shear phenomena, which we refer to as HShPs. We introduce bGO as

an exemplary material to enable the observation of these phenomena and experimentally demonstrate the symmetry-broken dispersion of the supported surface waves. The non-diagonalizable dielectric permittivity plays a key role in the unique properties of low-symmetry crystals, including monoclinic and triclinic lattices. Our findings are generalizable to engineered photonic systems with at least two non-orthogonal oscillators, including new metasurface designs capturing these physics. Beyond the results provided here for intrinsic, compensation-doped bGO, the presence of free charge carriers in bGO[32] may allow for methods for direct steering of the HShP propagation direction

(see Extended Data Fig. 6). Finally, exfoliation of thin flakes of single-crystal bGO has also been recently reported[33], which will allow to make use of volume-confined HShPs in such bGO thin films or — potentially — even in monolayers[34]. We anticipate that HShPs may have important implications in the manipulation of phase and directional energy transfer, including radiative heat transport[35], ultra-fast asymmetric thermal dissipation in the near field[35] and gate-tunability for on-chip all-optical circuitry[36]. Beyond advances in nanophotonics, infrared polariton propagation has been demonstrated as a means for quantifying crystal strain[37], polytypes[38], variations in free-carrier density, as well as phononic and electronic properties around defects[39], thereby also promising a new metrology tool for characterizing low-symmetry ultra-wide-bandgap semiconductors. We highlight that our results are applicable to any material with non-orthogonal optically active transitions and may therefore be extended to other optical phenomena, such as excitons in triclinic ReSe₂ (ref. [40]).

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

## Methods

### Experimental

The insulating (010)-oriented, $5 \times 5 \times 0.5$-mm$^3$ bGO substrate was produced by means of Fe compensation doping and was purchased from Novel Crystal Technology, Inc., Japan. The aQ sample was purchased from MaTeck GmbH, Germany. The absolute azimuth orientation of the samples was extracted from a global fit for each of the datasets of aQ and bGO (plotted in Fig. 2b,d, respectively), resulting in a rotation with respect to the principal $x$ axis of the laboratory coordinate system of $\Delta\phi_{bGO} = 27.95°$ and $\Delta\phi_{aQ} = 26.96°$. The aQ data have been rotated accordingly to shift the crystal axes (mirror planes) onto multiples of 90°. On the other hand, the bGO data are plotted as measured, as there is no principal azimuth angle for alignment because of the broken mirror symmetry. Here the simulation was rotated accordingly to match the data.

The Otto-type prism-coupling experiment measures the spectral dependence of surface waves through sharp absorption peaks observed as dips in the reflectance spectra, by using a prism placed near the material surface[28,41]. The crystals are oriented such that the monoclinic plane (bGO) and the optical axis (aQ) are parallel to the sample surface. By following the spectral position of the polariton resonances as a function of azimuth angle, we investigate the dispersion of hyperbolic waves at the surface for both bGO and aQ. The Otto geometry effectively selects a specific in-plane momentum component of those surface waves induced by the dipole excitation in Fig. 1e, f, as set by the incidence angle $\theta$ and azimuth angle $\phi$ that define the magnitude[28,29] and direction of the selected momentum, respectively.

As an excitation source for the Otto-type prism-coupled experiments, we use a mildly focused mid-infrared free-electron laser (FEL) (spot size ~1 mm$^2$) with small bandwidth (~0.3%) and wide tunability of 3–50 μm, covering the spectral range 350–800 cm$^{-1}$, in which aQ and bGO support polaritonic modes (details on the FEL have been reported elsewhere[42]). Although the frequency is scanned by tuning the FEL, different in-plane momenta can be accessed by means of changes in the incidence angle $\theta$ by rotating the entire Otto geometry (details on the setup have been reported elsewhere[28,43]). For the experiments shown in Fig. 2b, d, the incident angle was fixed to 28°, resulting in an in-plane momentum of $k_\parallel/k_0 \approx 1.10$ (at 500 cm$^{-1}$). In contrast to alternative approaches, the Otto geometry features experimental control over the excitation efficiency through tunability of the air gap width $d_{gap}$. Here the gap was adjusted to a separation in which all excited modes could be observed in the spectra simultaneously, that is, $d_{gap} \approx 8.3$ μm for bGO and $d_{gap} \approx 14.4$ μm for aQ. Direct readout of $d_{gap}$ with a range of 1–50 μm is realized through white-light interferometry, whereas the contrast of the interference range grants parallel alignment of prism and sample[43].

Mapping of the in-plane hyperbolic dispersion (Fig. 2f, h) was performed analogously to the Otto reflectance measurements shown in Fig. 2b, d. However, here we additionally varied the incidence angle $\theta$ to map out the frequency–momentum dispersion at each azimuth angle. Reflectance spectra were taken in a narrow frequency range of 670–730 cm$^{-1}$, at $\theta = 26°$, 28°, 30°, 32° and 34°, corresponding to in-plane momenta of $k_\parallel/k_0 \approx 1.03$, 1.11, 1.18, 1.25 and 1.32 (at 700 cm$^{-1}$), at nine azimuth angles. To allow prism coupling to the polaritons also for larger momenta, these data were taken at a constant air gap $d_{gap} \approx 4.0$ μm. The reflectance minima marking the polariton resonance were extracted from these data and are shown in Fig. 2f. The theoretical polariton resonance map, Fig. 2g, was calculated using a transfer matrix formalism[44], by extracting the peak positions of Im($r_{pp}$) of the air–bGO interface. To extract the single-frequency in-plane hyperbolic dispersion curves (Fig. 2h, i), we interpolated the momentum for a given frequency in the frequency–momentum dispersion for each measured/calculated azimuth angle.

### Theoretical

**Transfer matrix.** The calculations of the optical response shown in Fig. 2c, e and the polariton resonance map in Fig. 2g, as well as the dispersion maps in Extended Data Fig. 2c, d, were performed using a generalized $4 \times 4$ transfer matrix formalism[44]. In short, the formalism enables the calculation of reflection and transmission coefficients in any number of stratified media with arbitrary dielectric tensor, which enables to account for the anisotropy of our samples.

**COMSOL simulations.** COMSOL[45] version 5.6 was used for simulating point dipole excitation of HShPs on bGO. A point dipole was placed 100 nm above the surface of an infinite slab of bGO, with a dielectric permittivity matching that of ref. [6]. The dielectric function of aQ was taken from ref. [46]. Perfectly matched impedance boundary conditions were used on the sides of the simulation, which − in principle − absorb all radiation. However, to account for the imperfect behaviour of the boundaries, we ensured that the bGO slab was sufficiently large ($250 \times 250 \times 8$ μm), such that the wave is sufficiently damped when it reaches the boundary so as not to influence the results. Reciprocal space maps were generated by 2D Fourier transformation of the real-space electric field profiles.

**Isofrequency surface with complex frequency.** To obtain the isofrequency contour surface of the bulk wave when losses are considered in the materials, we turn to the complex-frequency method and solve the source-free Maxwell equations as follows:

$$\begin{pmatrix} \widetilde{\omega}^2\varepsilon_{xx} - \left(\widetilde{k}_z^2 + \widetilde{k}_y^2\right) & \widetilde{\omega}^2\varepsilon_{xy} + \widetilde{k}_x\widetilde{k}_y & \widetilde{k}_x\widetilde{k}_z \\ \widetilde{\omega}^2\varepsilon_{xy} + \widetilde{k}_x\widetilde{k}_y & \widetilde{\omega}^2\varepsilon_{yy} - \left(\widetilde{k}_z^2 + \widetilde{k}_x^2\right) & \widetilde{k}_y\widetilde{k}_z \\ \widetilde{k}_x\widetilde{k}_z & \widetilde{k}_y\widetilde{k}_z & \widetilde{\omega}^2\varepsilon_{zz} - \left(\widetilde{k}_y^2 + \widetilde{k}_x^2\right) \end{pmatrix}\begin{pmatrix} E_x \\ E_y \\ E_z \end{pmatrix} = 0,$$

in which $\varepsilon_{nn} = \varepsilon_{nn}^r + i\varepsilon_{nn}^k$, $nn = xx, yy, zz$ and $\varepsilon_{xy} = \varepsilon_{xy}^r + i\varepsilon_{xy}^k$. We fix the real component $k_0$ of the complex frequency $\omega$ and normalize it as $\widetilde{\omega} = \frac{\omega}{k_0} = 1 + i\widetilde{\omega}_i$, $\widetilde{\omega}^2 = 1 - \widetilde{\omega}_i^2 + 2i\widetilde{\omega}_i$, in which $\omega_i$ is an effective inverse mode lifetime, calculated neglecting the effect of the complex frequency on the material dispersion, and normalize the wavevector as $\widetilde{k}_{x,y,z} = \frac{k_{x,y,z}}{k_0}$. Note that we choose a negative sign for the time-dependent term e$^{-i\omega t}$, so $\widetilde{\omega}_i$ must be real and negative to reflect the decaying nature of the wave. The analytic expression $F\left(\widetilde{\omega}_i, \widetilde{k}_x, \widetilde{k}_y, \widetilde{k}_z\right) = 0$ is found by means of the secular equation of the above matrix, and two equations are obtained by separating the real and imaginary components of $F$, namely,

$$F_r\left(\widetilde{\omega}_i, \widetilde{k}_x, \widetilde{k}_y, \widetilde{k}_z\right) = 0,$$
$$F_i\left(\widetilde{\omega}_i, \widetilde{k}_x, \widetilde{k}_y, \widetilde{k}_z\right) = 0.$$

The isofrequency contour of the bulk wave and the imaginary component $\widetilde{\omega}_i$ are evaluated from those two equations. The numerical examples at 718 cm$^{-1}$ are given in Fig. 3g, h on the basis of the above method ($k_{x,y} \to k_{m,n}$). Notice that, in this procedure, we use the permittivity tensor elements calculated at Re($\omega$). When $\widetilde{k}_z = 0$, the analytic expression for the isofrequency contour of the bulk wave can be written as

$$\left[\widetilde{\omega}^2\left(\varepsilon_{xx}\varepsilon_{yy} - \varepsilon_{xy}^2\right) - \left(\varepsilon_{xx}\widetilde{k}_x^2 + \varepsilon_{yy}\widetilde{k}_y^2 + 2\varepsilon_{xy}\widetilde{k}_x\widetilde{k}_y\right)\right] = 0,$$

which turns into two equations by separating the real and imaginary parts,

$$\left(1 - \widetilde{\omega}_i^2\right)F_r - 2\widetilde{\omega}_i F_k - \left(\varepsilon_{xx}^r \widetilde{k}_x^2 + \varepsilon_{yy}^r \widetilde{k}_y^2 + 2\varepsilon_{xy}^r \widetilde{k}_x \widetilde{k}_y\right) = 0,$$

$$2\widetilde{\omega}_i F_r + \left(1 - \widetilde{\omega}_i^2\right)F_k - \left(\varepsilon_{xx}^k \widetilde{k}_x^2 + \varepsilon_{yy}^k \widetilde{k}_y^2 + 2\varepsilon_{xy}^k \widetilde{k}_x \widetilde{k}_y\right) = 0;$$

$$F_r = \varepsilon_{xx}^r \varepsilon_{yy}^r - \varepsilon_{xx}^k \varepsilon_{yy}^k - \left(\varepsilon_{xy}^{r\,2} - \varepsilon_{xy}^{k\,2}\right)$$

$$F_k = \varepsilon_{xx}^r \varepsilon_{yy}^k + \varepsilon_{xx}^k \varepsilon_{yy}^r - 2\varepsilon_{xy}^r \varepsilon_{xy}^k.$$

and they can be further simplified into the following equation

$$[(\varepsilon_{xx}^r F_k - \varepsilon_{xx}^k F_r)\widetilde{k}_x^2 + 2(\varepsilon_{xy}^r F_k - \varepsilon_{xy}^k F_r)\widetilde{k}_x \widetilde{k}_y + (\varepsilon_{yy}^r F_k - \varepsilon_{yy}^k F_r)\widetilde{k}_y^2]^2$$

$$+ 4(F_r^2 + F_k^2)[(\varepsilon_{xx}^r F_r + \varepsilon_{xx}^k F_k)\widetilde{k}_x^2 + 2(\varepsilon_{xy}^r F_r + \varepsilon_{xy}^k F_k)\widetilde{k}_x \widetilde{k}_y$$

$$+ (\varepsilon_{yy}^r F_r + \varepsilon_{yy}^k F_k)\widetilde{k}_y^2 - F_r^2 - F_k^2] = 0.$$

Therefore, the isofrequency curves in the $\widetilde{k}_z = 0$ plane as well as the imaginary component of the bulk complex frequency are obtained from the above expression.

## Characterization of the polariton modes in bGO

**Assignment of the polariton mode nature.** First, we briefly outline how conventional polariton materials are classified. A material supports surface polaritons at frequencies for which the real part of the crystal permittivity fulfils $\mathrm{Re}(\varepsilon) < -1$ (ref. [47]). In uniaxial crystals, the diagonal permittivity elements can be of different sign, leading to hyperbolic behaviour in which either the real part of one element is negative and the other two are positive (type I) or two are negative and one is positive (type II)[9]. However, this classification of anisotropic materials relies on the off-diagonal permittivity tensor elements being zero at all frequencies for an appropriate choice of coordinate system. The lower symmetry of bGO requires the emergence of $\varepsilon_{xy} \neq 0$. The dielectric permittivity elements using coordinates as indicated in Fig. 1a are shown in Extended Data Fig. 1a–e, but no coordinate system exists in which $\mathrm{Re}(\varepsilon_{xy}) \neq 0$ at all frequencies. This, as we have demonstrated in the Otto-geometry experiments (Fig. 2d), results in a non-trivial polaritonic response with highly directional modes that propagate along frequency-dictated propagation angles in the $a$–$c$ plane. Furthermore, because $\varepsilon_{xy} \neq 0$, it is not straightforward to determine whether the modes are elliptical ($\varepsilon_{xx}, \varepsilon_{yy}, \varepsilon_{zz} < -1$) or hyperbolic in nature (type I or type II), as the propagation angle is typically not aligned with one of the principal axes.

To unambiguously describe the nature of the polariton modes, we switch to a frequency-dispersive coordinate system [$mnz$], in which the real part of the permittivity tensor is diagonal. This is achieved by rotating the monoclinic plane by the frequency-dependent angle $\gamma(\omega)$ (equation (2)). The dispersion of $\gamma(\omega)$ and the resulting diagonal elements of $\varepsilon^{[mnz]}$ are plotted in Extended Data Fig. 1f–j. This new frequency-dispersive coordinate system enables the unique assignment of the supported polariton mode nature, which we have colour-coded in Extended Data Fig. 1g–j. For bGO, we observe the full range of possible combinations of positive and negative real parts of $\varepsilon_{mm}, \varepsilon_{nn}$ and $\varepsilon_{zz}$, leading to dielectric (white), elliptical (grey) and hyperbolic spectral regimes of type I (in-plane in blue, out-of-plane in red) and type II (in-plane in green, out-of-plane in yellow). By performing such a frequency-dependent rotation of the permittivity tensor, we have simplified the system into a pseudo-biaxial crystal at each frequency. However, as the dielectric tensor of a monoclinic crystal is not diagonalizable[20,48,49], the in-plane, off-diagonal element of $\varepsilon^{[mnz]}$ retains a non-vanishing imaginary part at all frequencies, that is, $\mathrm{Im}(\varepsilon_{mn}) \neq 0$ (plotted in Extended Data Fig. 1i), giving rise to the reduced symmetry of hyperbolic shear polaritons in monoclinic crystals, as discussed in Fig. 3.

The frequency-dependent rotation of the dielectric permittivity tensor is performed in three subsequent steps. First, the in-plane

permittivity tensor as shown in Extended Data Fig. 1a–c is rotated about the angle $\gamma(\omega)$ (equation (2)). However, $\gamma(\omega)$ has jumps of 90° at arbitrary frequencies, resulting in abrupt discontinuities in the real parts of $\varepsilon_{mm}$ and $\varepsilon_{nn}$, for which the two curves switch values. By analysing the derivative of $\varepsilon_{mm}$, we extract the frequency values $\omega_{\mathrm{jump}}$, in which the jumps occur and reassign the permittivity curves, respectively. At the eight in-plane TO frequencies $\omega_{\mathrm{TO}}$ of bGO, the permittivity curves feature a pole, which is also captured in the analysis of the derivative. Therefore, at this step, the resulting curves are smooth between the TO frequencies, but switch assignment at every $\omega_{\mathrm{TO}}$. The switching of the curves at every $\omega_{\mathrm{TO}}$ is performed in the last step. However, near $\omega_{\mathrm{TO}}$, the permittivity features a large imaginary part, which is not accounted for in the rotation angle $\gamma(\omega)$. This leads to poles in $\gamma(\omega)$ at $\omega_{\mathrm{TO}}$ (see Extended Data Fig. 1f), which – in turn – results in a small avoided crossing of $\varepsilon_{mm}$ and $\varepsilon_{nn}$ at the TO frequencies. Therefore, the reassignment of the last step results in discontinuous solutions near $\omega_{\mathrm{TO}}$, which is clearly not physical. To resolve this issue, we cut out ±1 cm$^{-1}$ in both $\varepsilon_{mm}$ and $\varepsilon_{nn}$ at all eight $\omega_{\mathrm{TO}}$ and smooth the curves by interpolation, resulting in the pseudo-biaxial permittivity curves as shown in Extended Data Fig. 1g–h. The rotation about $\gamma(\omega)$ also leads to discontinuities in the off-diagonal imaginary part $\mathrm{Im}(\varepsilon_{mn})$ at the frequencies $\omega_{\mathrm{jump}}$. However, because $\varepsilon_{mn} = \varepsilon_{nm}$, the abrupt rotation about 90° only leads to sign changes at every $\omega_{\mathrm{jump}}$. The curve of $\mathrm{Im}(\varepsilon_{mn})$ shown in Extended Data Fig. 1i is corrected for these sign changes.

**Polariton behaviour in bGO in the rotated frame.** To verify the polaritonic behaviour of bGO in the rotated frame, that is, the pseudo-biaxial crystal, we subsequently analyse the surface polariton dispersion in the rotated coordinate system [$mnz$] in Extended Data Fig. 2. For electric fields in the $m$–$z$ or $n$–$z$ planes, the analytical expression describing extraordinary surface polaritons in uniaxial crystals can be used[50]:

$$k_{\mathrm{polariton}} = \frac{\omega}{c}\sqrt{\frac{\varepsilon_{zz}\varepsilon_{\mathrm{in\text{-}plane}} - \varepsilon_{zz}}{\varepsilon_{zz}\varepsilon_{\mathrm{in\text{-}plane}} - 1}},$$

in which $\varepsilon_{\mathrm{in\text{-}plane}} = \varepsilon_{mm}, \varepsilon_{nn}$. In bGO, the solutions yield four polariton branches for each direction, $m$ and $n$, respectively, plotted as red dotted lines in Extended Data Fig. 2c, d. These analytical results are in perfect agreement with the numerically obtained surface polariton dispersion using a transfer matrix formalism[44]. To obtain the polariton propagation properties of the system, we calculate the full electric field patterns by placing a point dipole source above the bGO surface at $x = y = 0$ and simulating the optical response along the bGO–air interface ($z = 0$) with COMSOL Multiphysics[45]. The real-space field profiles clearly show the rotation of the major polarizability direction as a function of frequency, demonstrated for six different modes M1–M3 and N1–N3 (See Extended Data Fig. 2e–g, k–m). Frequencies are indicated as black dash-dotted lines in Extended Data Fig. 2a–d. Mode N4 is shown in Fig. 1e, g. The field profiles align with the rotated coordinate system, with basis vectors indicated by the '$m$' and '$n$' crosshair in each figure.

To relate the calculated dispersion of the polariton branches to the field profiles, we calculate the momentum–$k$ maps of these modes, as obtained by a 2D Fourier transformation of the respective electric field patterns of Extended Data Fig. 2e–g, k–m in Extended Data Fig. 2h–j, n–p, respectively. At all selected frequency positions, the electric field patterns contain a directional wave of large amplitude with low spatial frequency, as well as a wave with high spatial frequency. The observed in-plane momenta of the low-$k$ modes follow the modal dispersion predicted in Extended Data Fig. 2c, d, along the $m$ and $n$ axes for modes M1–M3 and N1–N3, respectively, as indicated by the black circles in Extended Data Fig. 2h–j, n–p. According to the mode characterization provided in Extended Data Fig. 1, these modes are hyperbolic, either of type I in-plane (M1–M3 and N1) or of type II in-plane (N2, N3). For all HShP modes, field patterns and $k$-space maps are characterized

by twofold rotational symmetry only, in agreement with the 2D plane group 2 (no mirror plane symmetries). Further, the distinct peaks in the $k$-space maps verify the principal polariton propagation direction, whereas the corresponding radial and azimuthal spreads are representative of their decay length and degree of directionality, respectively. Analogous to the model at 718 cm$^{-1}$ discussed in the main text, the maxima of the $k$-space maps shown in Extended Data Fig. 2 do not lie on the major polarizability axes (most prominently for the cases in Extended Data Fig. 2n, o), owing to shear phenomena in monoclinic bGO. This is discussed further in the main text and in Fig. 3.

## Data availability

The data and data analysis scripts that support the findings of this study are available from https://doi.org/10.5281/zenodo.5613335. Additionally, the Source data are provided with this paper.

## Code availability

The code generating Fig. 1c, d and Fig. 2g, i, j can be downloaded from https://doi.org/10.5281/zenodo.5613335. The code used for Fig. 3g–i is available from Andrea Alù (aalu@gc.cuny.edu) on reasonable request.

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

**Acknowledgements** We thank W. Schöllkopf and S. Gewinner (FHI Berlin) for operating the infrared free-electron laser. N.C.P., G.C., M.W. and A.P. thank R. Ernstorfer (TU Berlin) and S. Mährlein (FHI Berlin) for careful reading of the manuscript. N.C.P. acknowledges support by the International Max Planck Research School for Functional Interfaces in Physics and Chemistry. G.H. acknowledges the support from A*STAR AME Young Individual Research Grants (YIRG, no. A2084c0172). X.N., G.H. and A.A. acknowledge the Office of Naval Research with grant no. N00014-19-1-2011 and the Vannevar Bush Faculty Fellowship. M.S. acknowledges National Science Foundation awards DMR 1808715, DMR 1420645 and OIA-2044049, and Air Force Office of Scientific Research awards FA9550-18-1-0360, FA9550-19-S-0003 and FA9550-21-1-0259, and the Knut and Alice Wallenberg Foundation. J.D.C. was supported by the Office of Naval Research grant no. N00014-18-12107 and J.R.M. by the National Science Foundation, Division of Materials Research grant no. 1904793. T.G.F. was supported by University of Iowa Startup Funding.

**Author contributions** N.C.P. and G.C. performed the experiments. N.C.P. and A.P. performed the transfer matrix simulations. N.C.P., T.G.F. and A.P. performed COMSOL simulations. X.N., G.H. and A.A. performed the analytical derivations and the calculation of isofrequency surfaces. All authors contributed to writing the manuscript. A.A., J.D.C., T.G.F. and A.P. oversaw the project. M.S. and J.D.C. initiated the research.

**Funding** Open access funding provided by Max Planck Society.

**Competing interests** The authors declare no competing interests.

**Additional information**
**Correspondence and requests for materials** should be addressed to Andrea Alù, Joshua D. Caldwell, Thomas G. Folland or Alexander Paarmann.

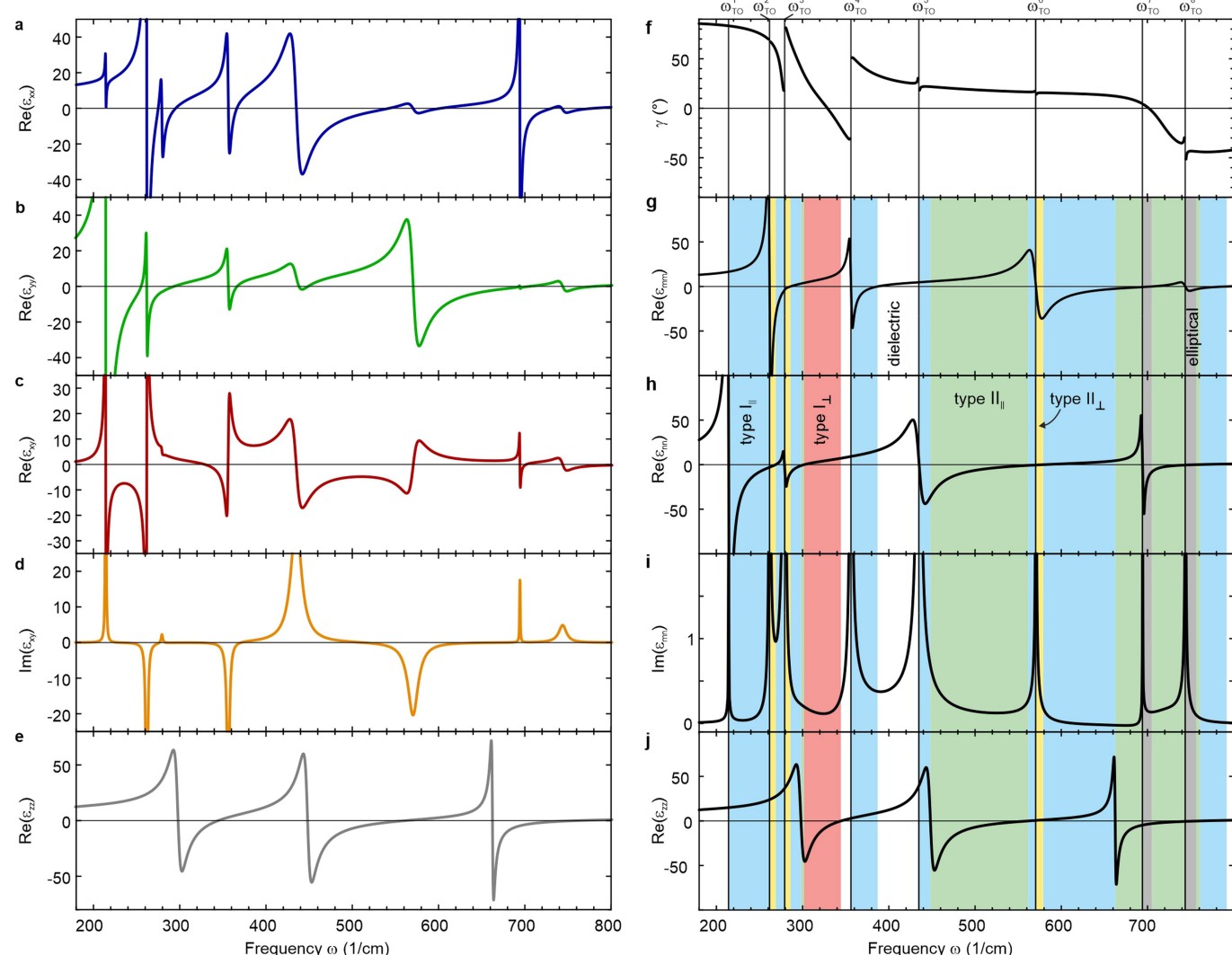

**Extended Data Fig. 1 | Frequency-dispersive dielectric permittivity of bGO and surface polariton mode assignment. a–e**, Dielectric permittivity tensor elements $\varepsilon_{xx}$, $\varepsilon_{yy}$, $\varepsilon_{xy}$ ($=\varepsilon_{yx}$) and $\varepsilon_{zz}$, respectively, of charge carrier-free bGO at infrared frequencies[6]. **f**, Rotation angle $\gamma$ (equation (2)) as a function of frequency. **g**, **h**, Diagonal elements $\varepsilon_{mm}$ and $\varepsilon_{nn}$ of the frequency-dispersive in-plane permittivity, featuring four distinct reststrahlen bands each.

**i**, Non-zero imaginary part of the off-diagonal element $\text{Im}(\varepsilon_{mn})$. **j**, Unchanged out-of-plane permittivity $\varepsilon_{zz}$. Depending on the combination of positive or negative real parts of $\varepsilon_{mn}$, $\varepsilon_{nn}$ and $\varepsilon_{zz}$, different types of phonon polariton are supported (colour-shaded), such as elliptical SPhPs and type I and type II, in-plane (∥) and out-of-plane (⊥) hyperbolic polariton modes, respectively.

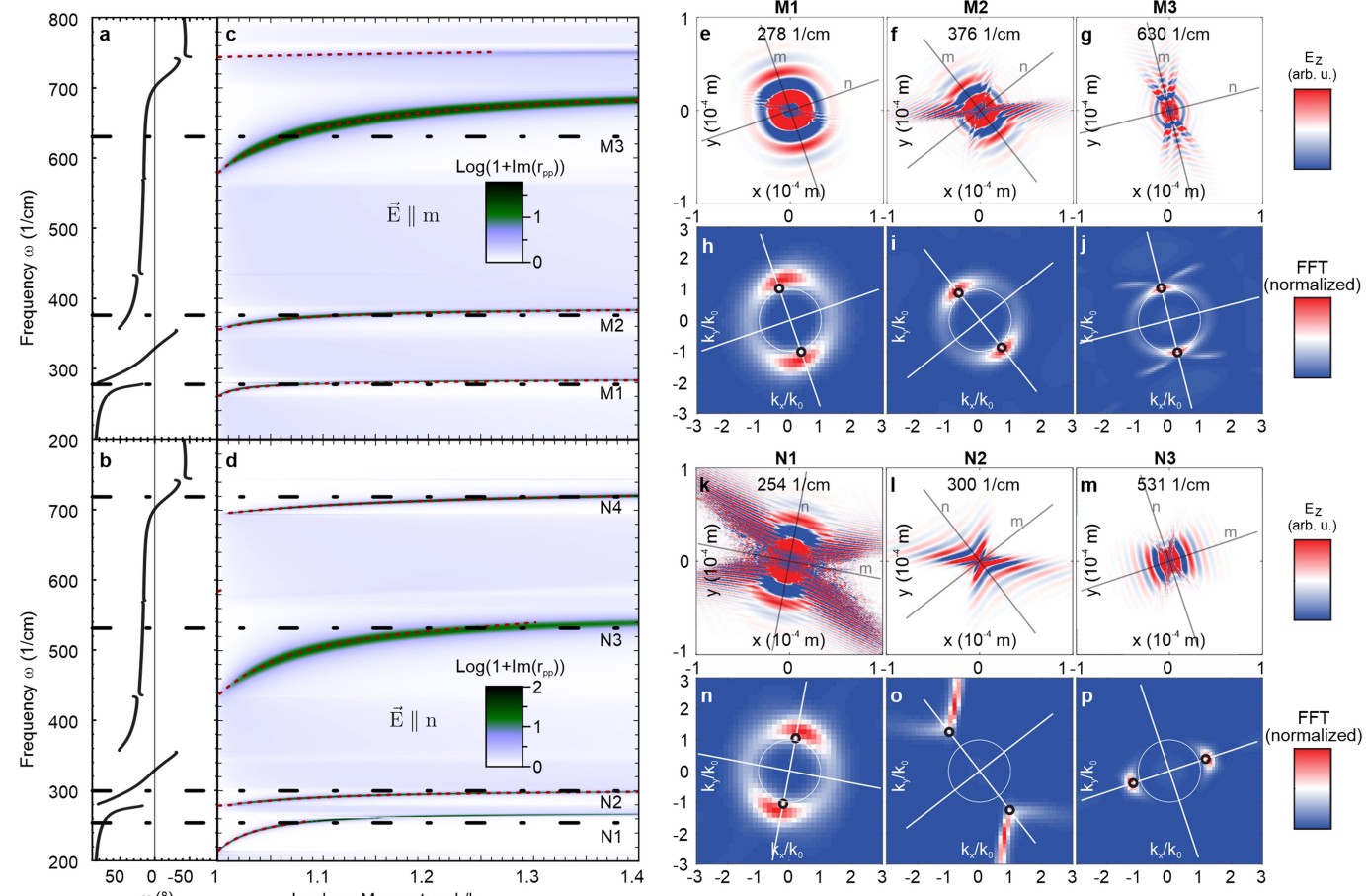

**Extended Data Fig. 2 | Rotating HShPs on bGO. a, b**, Rotation angle γ as a function of frequency. **c, d**, Dispersion of HShPs on the surface of a bulk bGO crystal calculated with the frequency-dispersive permittivity tensor along the (rotating) *m* axis (**c**) and the *n* axis (**d**), obtained using a transfer matrix method[44]. The four supported polaritons along each axis *m* and *n* are clearly distinguishable, in perfect agreement with the theoretically calculated polariton dispersion (red dotted lines)[50]. Black horizontal dash-dotted lines mark the frequencies M1–M3 and N1–N4 at which the electric field distribution is plotted. N4 is shown in Fig. 1e, g. **e–g**, Real-space electric fields at the bGO surface at frequencies M1–M3, respectively. **h–j**, The respective

two-dimensional Fourier transformation. The fields were calculated using COMSOL Multiphysics[45] (see Methods for details). **k–m**, Real-space electric fields at frequencies N1–N3, respectively. **n–p**, The respective Fourier transforms. All maps (**e–p**) were calculated using the non-dispersive permittivity tensor (Extended Data Fig. 1a–e), thus showing rotated field patterns with different orientations, depending on the frequency. The thin black and white crosshairs indicate the principal axes of the respective frequency-dispersive coordinate system, its rotation given by γ at the corresponding frequency. Small black circles in **h–j** and **n–p** mark the momentum value of the analytical dispersion in **c** and **d**, respectively.

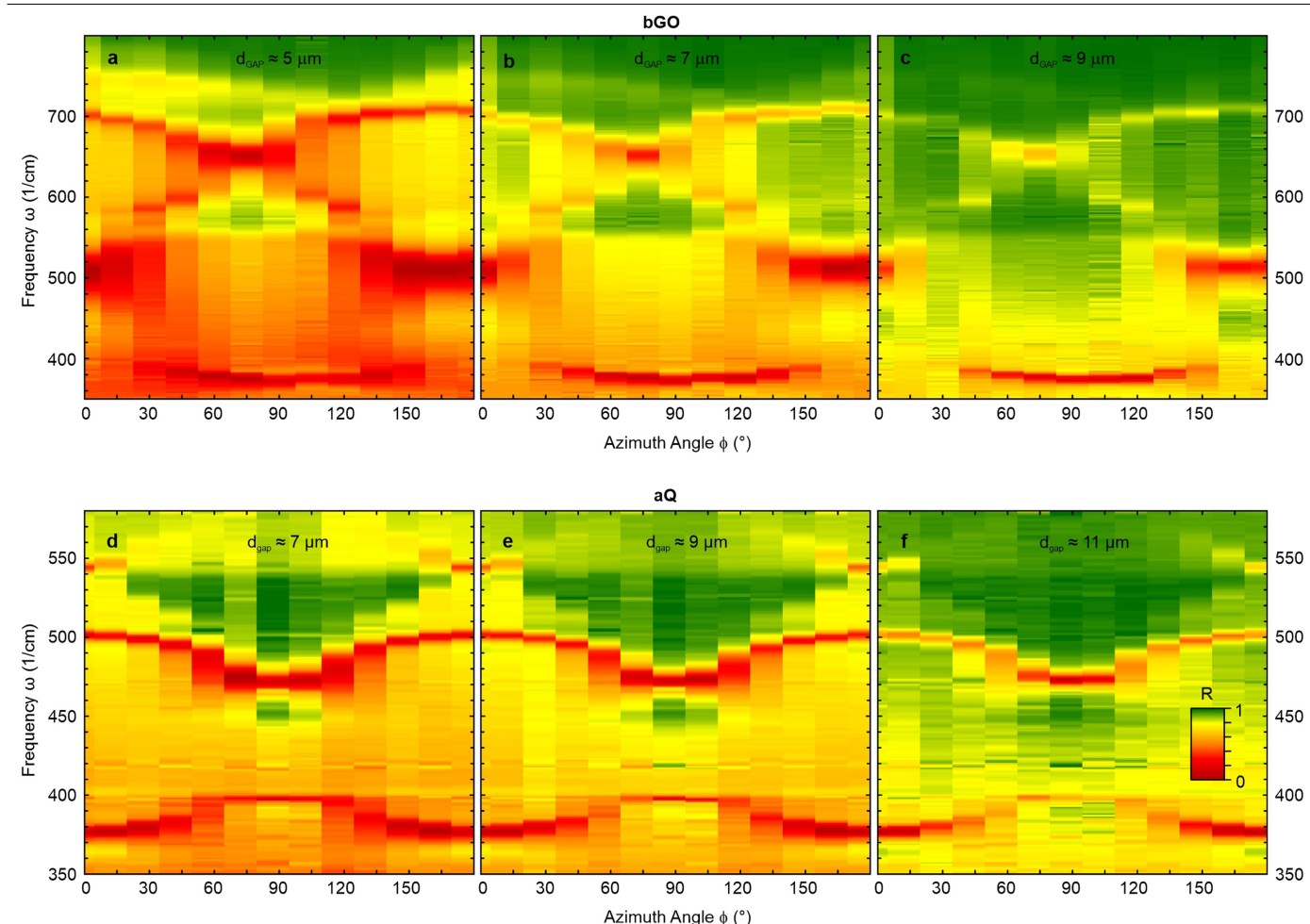

**Extended Data Fig. 3 | Experimental datasets for bGO and aQ at different gap sizes.** The gap size $d_{gap}$ in our Otto geometry setup can be tuned and monitored[43], enabling control over the excitation efficiency of the polariton modes[44]. Datasets measured for bGO (**a**–**c**) and for aQ (**d**–**f**) at three different gap sizes each. For smaller gaps, some modes are overcoupled and their resonance features broadened (such as the mode at 500 cm⁻¹ in bGO), whereas for larger gaps, some modes are undercoupled and their resonance features too weak to be clearly distinguishable (in particular, the mode at 725 cm⁻¹ in bGO). The centre gap sizes compromise between these effects. Note that the gap sizes indicated here are the values monitored with a white-light interferometry setup[43]. The fits performed for the datasets shown in Fig. 2, that is, the datasets shown in Extended Data Fig. 3b for bGO and Extended Data Fig. 3e for aQ, however, yielded larger gap sizes of 8.3 μm (bGO) and 10.4 μm (aQ). The offset can be attributed to non-perfect parallel alignment between prism and sample and a lateral offset between the polariton excitation site with the FEL and the white-light spot for the gap measurement.

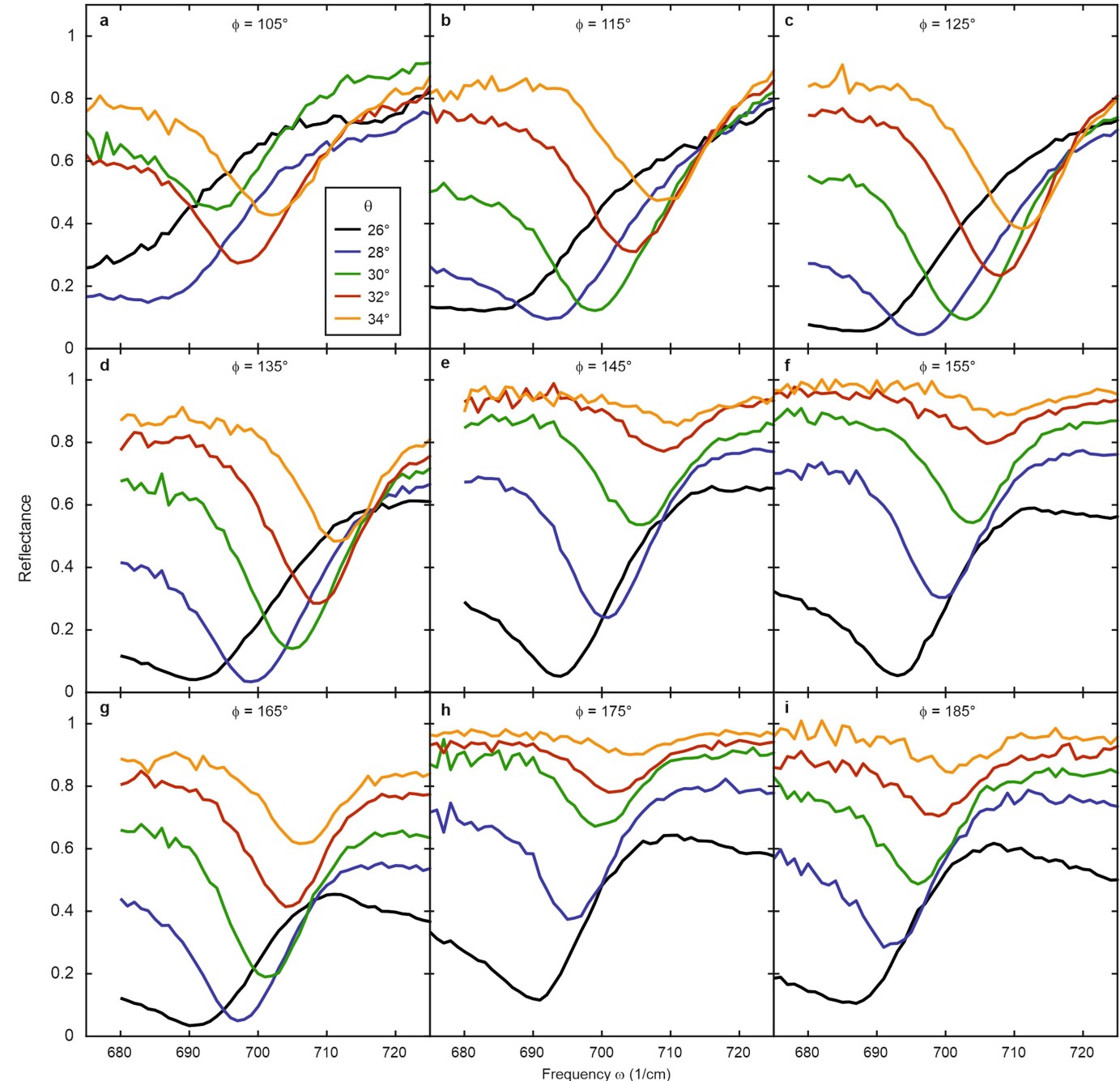

**Extended Data Fig. 4 | Mapping of the in-plane hyperbolic dispersion in bGO.** Experimental Otto reflectance spectra for various azimuthal angles (**a**–**i**), each at five different incidence angles $\theta = 26°, 28°, 30°, 32°$ and $34°$. All data were acquired at constant Otto air gap of $d_{gap} \approx 4.0$ μm. Further data were taken also for $d_{gap} \approx 6.0$ μm but showed the same behaviour at generally reduced polariton resonance amplitudes.

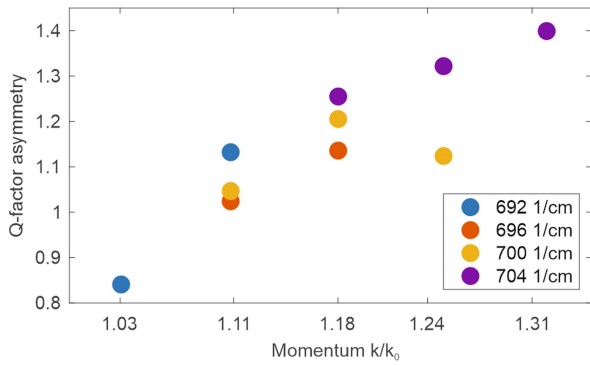

**Extended Data Fig. 5 | Q-factor asymmetry along the in-plane hyperbolic dispersion of bGO.** The full width at half maximum (FWHM) of the polariton resonance dips in Extended Data Fig. 4 was extracted, indicative of the (inverse) quality factor (Q-factor) of the polariton. We used the MATLAB function 'findpeaks' to extract the FWHM of the resonance dips, which works robustly independent of the specific peak shapes. To account for the effect of different optical coupling efficiencies for different momenta in the Otto geometry at constant air gap[28], the FWHM should only be compared at the same momentum values. Therefore, we first interpolated the azimuth angle at which the polariton crosses the experimental momenta from Fig. 2f and then interpolated the resonance width extracted for these momenta from Extended Data Fig. 4 to that same azimuth angle. This procedure was performed for each frequency at both sides of the hyperbolic curves (Fig. 2h). Finally, the Q-factor asymmetry is given as the ratio of the FWHM on the left side (smaller azimuth angles) and the right side (larger azimuth angles) of the hyperbolic dispersion for each momentum and frequency. Please note that not all hyperbolic dispersions in Fig. 2h cross all experimental momentum values, which is why the Q-factor asymmetry can only be evaluated at selected momentum values as shown for each frequency. Owing to the larger experimental uncertainty of the FWHM analysis (as compared with the peak position), we refrained from extrapolation during this procedure. Also note that the polariton dip shapes for $k/k_0 = 1.03$ (incidence angle $\theta = 26°$) are strongly asymmetric (see Extended Data Fig. 4), such that the quality factor analysis is not very reliable there, explaining the outlier for which the Q-factor asymmetry is <1. For all other cases, we find values clearly >1, that is, we find consistently that the Q-factor is smaller on the left side than on the right side of the hyperbolic dispersion. Also, the data show a trend of increasing asymmetry with increasing momenta, that is, further out in the dispersion. Both effects are in very good agreement with the theoretical predictions (Fig. 3).

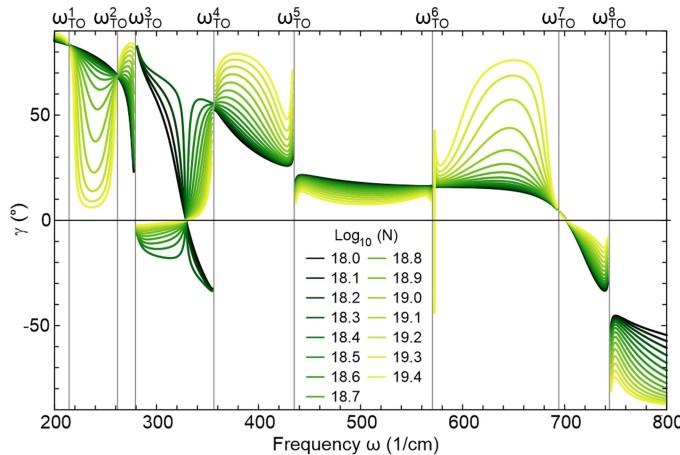

**Extended Data Fig. 6 | Active tuning of the propagation direction of HShPs in bGO.** Rotation angle γ (equation (2)) calculated as a function of doping concentration N (in cm⁻³), assuming a Drude contribution with anisotropic charge carrier mobility, $\mu_x = 296\ cm^2\ V^{-1}\ s^{-1}$, $\mu_y = \mu_z = 37\ cm^2\ V^{-1}\ s^{-1}$. (Literature values of $\mu$ in bGO feature a large variance[51]. We assume here strong anisotropy of the charge carrier mobility to emphasize the rotation mechanism.) Clearly, between the TO frequencies at which the HShPs disperse, the rotation angle γ is strongly dependent on the doping concentration, enabling active tuning of the propagation direction of the supported polariton modes. Note that, for an isotropic Drude contribution, on the other hand, equation (2) predicts no rotation of the propagation direction as a function of doping concentration.

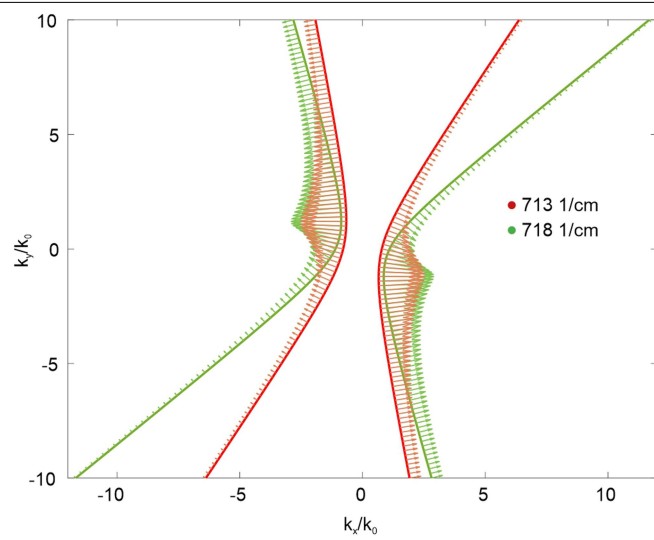

**Extended Data Fig. 7 | Group velocity analysis for hyperbolic shear polaritons.** Analytical solutions of isofrequency contours of lossless bGO for $k_z = 0$ (see Fig. 1c) are used to numerically evaluate gradients with respect to frequency along the curves. The resulting group velocity ($v_g$) distribution is shown as arrows along the isofrequency curves. The maximum value near the base of the hyperbola for $\omega = 718$ cm$^{-1}$ is $|v_g| \approx 0.038c_0$, in which $c_0$ is the speed of light in vacuum. The continuous rotation of the hyperbola axis with frequency (see main text) results in a pronounced asymmetry of the group velocity distribution: the peak of $|v_g|$ is shifted off the base of the hyperbola (**a**) and $|v_g|$ is notably different along the two arms of the hyperbola (**b**). This asymmetric distribution of $|v_g|$ certainly contributes greatly to the tilted wavefronts of HShPs observed in the main text. The asymmetric distribution of $|v_g|$ around the base of the hyperbola is in excellent agreement with the results shown in Fig. 2j.