## [Peer Review File · Nature]

Manuscript Title: Hyperbolic Shear Polaritons in Low-Symmetry Crystals

Reviewer Comments & Author Rebuttals

Reviewer Reports on the Initial Version:

Referee #1 (Remarks to the Author):

Dear authors, I appreciate the opportunity of reviewing your work.

The work introduces shear polaritons as a new modality of these quasiparticles. It claims that new polaritonics features can emerge from bGO which are related to its monoclinic crystal structure. Continuous steering of propagation direction by frequency tuning, tilted wavefronts and asymmetric dielectrics are among the properties brought by this class of crystals, which are complementary to reported orthorhombic and hexagonal nanophotonic crystals. Therefore, it is claimed here that the discoveries presented can boost the polaritons physics field regarding low-symmetry materials, expanding the possibilities for devices developments.

The manuscript is extremely well organized and present a solid description of the formalism or concept that justify the existence of the named hyperbolic shear polaritons. I particularly enjoyed the reading, and the narrative carries a comparison between bGO and aQ. The symmetry break is unquestionable specially regarding the 2D FT maps. In spite of the robust conceptual description based on numerical and theoretical results, the only experimental evidence presented is an energy-angle map measured using evanescent waves through a prism in the Otto-type experiment illuminated by FEL radiation. The mid to far-IR range covered comprised the HShPs predicted by simulations and theory.

I appreciate the results presented and recognize its scientific merit, however, as a potential complement to other important advances in hyperbolic polaritonics, the work doesn't follow the same route on showing direct observation of the waves from where, usually, fundamental properties of these polaritons are retrieved.

Therefore, I would revisit my assessment and potentially consider the publication in Nature if more direct experimental evidence of these waves is brought in.

Below I list my concerns:

1. As far as I could find in the manuscript (and SI), there is not many details on the sample's geometry. I understand that the study approaches surface waves, however, is there any contribution from crystals edges in the Otto experiment? In other words, is the sample large enough for edges waves (either edge launching or edge reflection) to be neglected? I recommend a more detailed experimental description as this is the only experiment presented. A scheme showing the probe size and the crystal in good proportion would help making this discussion promptly accessible.
2. In many reported works, numerical simulations considering a dipole as a polariton source represent, with good accuracy, the point source/probe of experimental modalities such as s-SNOM. How can an Otto configuration that excites multiple surface modes can be so precisely described by a point dipole simulation? I honestly missed a discussion with more details of the numerical simulation where the distance dipole-surface is defined and how modes excitation matches the modes selected by the prism-sample gap.
3. In a more general perspective, the work is a prime step towards complementing the catalog of

crystal types for hyperbolic polaritonics. In the theme of anisotropic or birefringent media for volume waves confinement, works introducing these waves in hBN [1] and MoO₃ [2] for instance were strongly supported by direct observation of those waves by near-field nanoscopy. I would expect that extending this knowledge to low-symmetry crystals would require similar experimental robustness in case the authors aim to set these findings in the same class of the mentioned works.

I totally understand that this energy range is only accessible to large accelerators such as FEL and Synchrotrons. This brings one more uniqueness factor to the discoveries presented and also emphasizes how challenging is to access those ranges. Recent reports [3-5] on s-SNOM operating in the mid to far-IR range can be of inspiration for the authors to pursue such additional evidence of these waves.

References:

[1] S. Dai, Z. Fei, Q. Ma, A. S. Rodin, M. Wagner, A. S. McLeod, M. K. Liu, W. Gannett, W. Regan, K. Watanabe, T. Taniguchi, M. Thiemens, G. Dominguez, A. H. C. Neto, A. Zettl, F. Keilmann, P. Jarillo-Herrero, M. M. Fogler, D. N. Basov, *Science* (80-.). 2014, 343, 1125.

[2] W. Ma, P. Alonso-González, S. Li, A. Y. Nikitin, J. Yuan, J. Martín-Sánchez, J. Taboada-Gutiérrez, I. Amenabar, P. Li, S. Vélez, C. Tollan, Z. Dai, Y. Zhang, S. Sriram, K. Kalantar-Zadeh, S.-T. Lee, R. Hillenbrand, Q. Bao, *Nature* 2018, 562, 557.

[3] F. H. Feres, R. A. Mayer, L. Wehmeier, F. C. B. Maia, E. R. Viana, A. Malachias, H. A. Bechtel, J. M. Klopff, L. M. Eng, S. C. Kehr, J. C. González, R. O. Freitas, I. D. Barcelos, *Nat. Commun.* 2021, 12, 1995.

[4] T. V. A. G. Oliveira, T. Nörenberg, G. Álvarez-Pérez, L. Wehmeier, J. Taboada-Gutiérrez, M. Obst, F. Hempel, E. J. H. Lee, J. M. Klopff, I. Errea, A. Y. Nikitin, S. C. Kehr, P. Alonso-González, L. M. Eng, *Adv. Mater.* 2021, 33, 2005777.

[5] I. D. Barcelos, T. A. Canassa, R. A. Mayer, F. H. Feres, E. G. de Oliveira, A.-M. B. Goncalves, H. A. Bechtel, R. O. Freitas, F. C. B. Maia, D. C. B. Alves, 2021, arXiv:2105.01148

Referee #2 (Remarks to the Author):

The paper studies the propagation of phonon polaritons in crystals with low symmetry, and identifies a few interesting properties of surface phonon polaritons in this materials. While the "non-diagonalizability" of the permittivity tensor is already well known (including its consequence on the plane waves in the bulk of the material), the original contribution of the paper is to study the new class of polaritons in these materials, and the properties of the supported surface waves, with a theoretical and experimental study. Previous literature is adequately cited and discussed. The paper is technically correct (except from a few minor issues listed below), the experimental method is sound, I would optionally improve the theoretical part (see minor issues below). The results are clearly explained and described

This is an interesting and an overall high quality contribution, expanding our understanding of phonon polaritons. The results are important for the photonics community, but my main concern is that the original content of this paper and the observed phenomena might not be of sufficient interest for a broad scientific community, and it is probably more suitable for a specialized journal such as *Nature Photonics* or *Nature Physics*, in which I would fully support its publication once the minor issues have been solved.

Minor issues:

-One minor issue is the use of the word "diagonalizable": the Authors say multiple times that the dielectric tensor cannot be diagonalized. This is not entirely accurate, because matrices in the form of equation 1 can be diagonalized on a complex field in general, meaning that it is possible to find V and D such that $V^*D^*V^{-1}$ is equal to the original matrix. The special property here is that for the tensors considered in this paper V cannot be a real constant matrix (corresponding to a real rotation of the axes). Instead it has to be complex and frequency dependent. That implies that there is no system of coordinates (axes) that can be chosen so that both the real and the imaginary part of the matrix are diagonal. This point should be clarified in the main text, so it is clearer what it is meant by "non-diagonalizable"

-Page 4: This sentence "In addition, the wavefronts are tilted with respect to the direction of energy flow, with no apparent mirror symmetry. This feature can also be clearly seen by examining the Fourier transform of the real-space profile (Fig. 1g), exhibiting a stronger intensity along one side

130 of the hyperbola" is, I believe, not completely justified: it makes sense that the direction of the phase fronts is different from the energy flow, but figure 1g is not sufficient to justify this claim, since it is provided for a single frequency. A complete justification requires the calculation of the group velocity (which would be a very interesting addition to the study), which is done taking a differential in the frequency, so it requires calculation in at least two frequencies close to each other. Otherwise, only the phase velocity can be calculated. The asymmetry mentioned in this sentence arises from the fact that, considering a wave in a specific direction, the k vector has different attenuation (imaginary part) in the various quadrants identified by the hyperbola in the real part of k , as in figure 3. It would be beneficial and much more rigorous to plot real and imaginary part of k separately after solving Maxwells equation as opposed to the Fourier transform of the numerical full field simulations.

Referee #3 (Remarks to the Author):

In this work, Passler and co-authors describe theoretical and experimental work on identifying "hyperbolic shear polaritons" which they state can exist in low-symmetry crystals due to the presence of non-orthogonal IR-active phonon resonances. The work is novel and timely, and the manuscript appears to be well organized and well thought-out. I have provided some queries below that I had while reading the work, although generally I believe this work is well-suited for publication in Nature after revisions.

1. In describing the complex dielectric tensor for the materials discussed, ("bGO" and "aQ"), the authors give a general form in Eq. 1. This is fine, however I am wondering what actually was used to compute these frequency-dependent terms. They presumably come from information related to the phonon eigenvectors and eigenenergies, which I imagine can be calculated using a first principles (ab initio) approach. Can the authors comment on how they arrived at their values for $\epsilon(\omega)$?

2. There are several debates around phonon-polaritons in monolayers (for example see Nano Lett. 2019, 19, 4, 2653–2660 and Nano Lett. 2017, 17, 6, 3758–3763). Do any of the considerations presented there appear in the case of hyperbolic shear polaritons?

3. In describing this $\epsilon(\omega)$, the authors state that the off-diagonal terms $\epsilon_{xy}(\omega)$ and $\epsilon_{yx}(\omega)$ would be the same due to reciprocity. Later the authors state that $\text{Re}[\epsilon(\omega)]$ is Hermitian. What confuses me is then the final paragraph on page 7. At what point do things become "non-Hermitian", as the authors suggest? Can the authors clarify what is meant by that, and whether or not the systems studied in this work are considered non-Hermitian? Although the paragraph introduces interesting

ideas, I found it a bit confusing in terms of how it connected with the present work directly, so perhaps readers would appreciate a bit of a clearer connection.

Author Rebuttals to Initial Comments:

Referee #1 (Remarks to the Author):

Dear authors, I appreciate the opportunity of reviewing your work.

The work introduces shear polaritons as a new modality of these quasiparticles. It claims that new polaritonic features can emerge from bGO which are related to its monoclinic crystal structure. Continuous steering of propagation direction by frequency tuning, tilted wavefronts and asymmetric dielectrics are among the properties brought by this class of crystals, which are complementary to reported orthorhombic and hexagonal nanophotonic crystals. Therefore, it is claimed here that the discoveries presented can boost the polaritons physics field regarding low-symmetry materials, expanding the possibilities for devices developments.

The manuscript is extremely well organized and present a solid description of the formalism or concept that justify the existence of the named hyperbolic shear polaritons. I particularly enjoyed the reading, and the narrative carries a comparison between bGO and aQ. The symmetry break is unquestionable specially regarding the 2D FT maps. In spite of the robust conceptual description based on numerical and theoretical results, the only experimental evidence presented is an energy-angle map measured using evanescent waves through a prism in the Otto-type experiment illuminated by FEL radiation. The mid to far-IR range covered comprised the HShPs predicted by simulations and theory.

I appreciate the results presented and recognize its scientific merit, however, as a potential complement to other important advances in hyperbolic polaritronics, the work doesn't follow the same route on showing direct observation of the waves from where, usually, fundamental properties of these polaritons are retrieved.

We thank the reviewer for the positive assessment of our work regarding the importance, novelty and potential impact. In response to the reviewers request for a more direct experimental evidence of the hyperbolic shear polaritons, we went back to the lab to acquire a significant amount of additional data that now allows us to provide direct experimental proof of the hyperbolic dispersion of the surface waves in bGO, now included in the revised manuscript. In these experimental data, we also find clear indication of the shear effect by observing consistent asymmetries in the quality factors along both arms of the hyperbolic dispersion.

Therefore, I would revisit my assessment and potentially consider the publication in Nature if more direct experimental evidence of these waves is brought in.

In the revised manuscript, we now provide this direct experimental evidence. Thus, we are convinced that the reviewer may now reconsider this assessment.

Below I list my concerns:

1. As far as I could find in the manuscript (and SI), there is not many details on the sample's geometry. I understand that the study approaches surface waves, however, is there any contribution

from crystals edges in the Otto experiment? In other words, is the sample large enough for edge waves (either edge launching or edge reflection) to be neglected? I recommend a more detailed experimental description as this is the only experiment presented. A scheme showing the probe size and the crystal in good proportion would help making this discussion promptly accessible.

We thank the reviewer for this comment. The sample is a commercially diced $5 \times 5 \times 0.5 \text{ mm}^3$ bulk piece, and the mildly focused FEL beam has a spot size of $\sim 1 \text{ mm}$ at the sample which we make sure to hit centrally and not at the edges. So, no, there are no contributions from edges and, yes, the sample is big enough. The reviewer is certainly thinking of similar experiments on exfoliated (small) flakes of 2D materials, where indeed an Otto-type measurement with this setup would be quite challenging. For large bulk crystals, however, this is not an issue at all. We added the information on sample size and laser spot size to the methods section.

2. In many reported works, numerical simulations considering a dipole as a polariton source represent, with good accuracy, the point source/probe of experimental modalities such as s-SNOM. How can an Otto configuration that excites multiple surface modes can be so precisely described by a point dipole simulation? I honestly missed a discussion with more details of the numerical simulation where the distance dipole-surface is defined and how modes excitation matches the modes selected by the prism-sample gap.

We thank the reviewer very much for this question. To resolve the comment, we kindly note the conceptual difference between sSNOM and Otto, which in a way correspond to Fourier transformations of each other: while sSNOM excites (and probes) a point in real space and (ideally) addresses all momenta, Otto specifically excites a point in momentum space but it averages in real space. The momentum selection arises from the incidence angle, which sets the momentum magnitude, and the sample azimuthal orientation with regard to the incidence plane in the Otto geometry. What this means is that a single Otto measurement represents a specific point in the 2D momentum map. This is in contrast to a sSNOM measurement or a simulated point-dipole excitation, where the data is acquired in real space and the momentum information (such as the hyperbolic dispersion) can only be accessed using Fourier analysis.

Thus, both approaches allow accessing the properties of surface waves, though in different ways. In our work, we do not directly use the point-dipole simulations to predict the Otto measurements, nor vice-versa. However, both approaches probe the same modes. We believe that, with the additional experimental data provided in the revised version of the manuscript, this also becomes clear in the manuscript now.

For spatially homogeneous surfaces like in our study, the reviewer's statement "How can an Otto configuration that excites multiple surface modes [...]" is not correct: an sSNOM measurement is prone to excite multiple surface modes simultaneously, while the Otto scheme is fully selective in momentum space. Of course for any kind of nanostructured surface, e.g., multiple exfoliated flakes of dimensions comparable to the wavelength, the situation reverses, and indeed Otto may (spatially) excite multiple surface modes while sSNOM could become (spatially) more selective.

Further, the prism-sample gap in the Otto scheme does not select the momentum but merely changes the efficiency of polariton excitation, as discussed in some detail in Ref. 43 of the manuscript. There is no direct relation between the dipole-surface gap in the simulations, Fig. 1 and 3, and the prism-

sample gap for the Otto measurements in Fig. 2. Both are simply parameters to steer the efficiency of optical interaction between the excitation source and the evanescent surface waves. Please also note that theoretical support for the Otto measurements is provided by the transfer matrix calculations and not (directly) by the point dipole simulations.

3. In a more general perspective, the work is a prime step towards complementing the catalog of crystal types for hyperbolic polaritonics. In the theme of anisotropic or birefringent media for volume waves confinement, works introducing these waves in hBN [1] and MoO₃ [2] for instance were strongly supported by direct observation of those waves by near-field nanoscopy. I would expect that extending this knowledge to low-symmetry crystals would require similar experimental robustness in case the authors aim to set these findings in the same class of the mentioned works.

I totally understand that this energy range is only accessible to large accelerators such as FEL and Synchrotrons. This brings one more uniqueness factor to the discoveries presented and also emphasizes how challenging is to access those ranges. Recent reports [3-5] on s-SNOM operating in the mid to far-IR range can be of inspiration for the authors to pursue such additional evidence of these waves.

We sincerely thank the reviewer for requesting more direct experimental evidence of the hyperbolic shear modes which inspired us to do experiments that we had previously thought unfeasible. We are certain that these additional data significantly strengthen our paper, which we would have missed without the reviewer's comments.

On the other hand, we kindly ask the reviewer to acknowledge the power of the Otto experiments, in particular for measuring the polariton dispersion. In fact, since the Otto geometry measures directly in momentum space, rather than sSNOM which requires FFTs to access the dispersion, we are convinced that our results now provide the most direct experimental evidence for hyperbolic shear waves. This holds true in particular for (comparably) low-momentum surface waves of bulk crystals investigated in our work, in contrast to high-momentum volume-confined modes in ultrathin films.

We also would like to mention that in other disciplines dispersion measurements are routinely done directly in momentum space, for instance for the dispersion of electronic states using angular-resolved photoemission (ARPES). In this sense, we show now in our paper for the first time that momentum-space techniques can access in-plane dispersion with high precision also for polariton modes. Once again, we are very grateful to the reviewer because without this comment we would not have dared to push that far, leading to a significant improvement of our paper.

Referee #2 (Remarks to the Author):

The paper studies the propagation of phonon polaritons in crystals with low symmetry, and identifies a few interesting properties of surface phonon polaritons in these materials. While the “non-diagonalizability” of the permittivity tensor is already well known (including its consequence on the plane waves in the bulk of the material), the original contribution of the paper is to study the new class of polaritons in these materials, and the properties of the supported surface waves, with a theoretical and experimental study. Previous literature is adequately cited and discussed. The paper is technically correct (except from a few minor issues listed below), the experimental method is sound, I would optionally improve the theoretical part (see minor issues below). The results are clearly explained and described

This is an interesting and an overall high quality contribution, expanding our understanding of phonon polaritons. The results are important for the photonics community, but my main concern is that the original content of this paper and the observed phenomena might not be of sufficient interest for a broad scientific community, and it is probably more suitable for a specialized journal such as *Nature Photonics* or *Nature Physics*, in which I would fully support its publication once the minor issues have been solved.

We thank the reviewer for the positive assessment of the quality and novelty of our work. We have now expanded the original content of our work in the revised manuscript, in which we now also map out the in-plane hyperbolic dispersion of the polaritons with the Otto geometry. Apart from the added scientific benefit, we now additionally provide a novel methodology to study in-plane hyperbolic modes, which has not been reported in the literature prior to our work. We are thus convinced that our revised manuscript now meets the stringent publication criteria of *Nature*. In the following, we address the additional points raised.

Minor issues:

-One minor issue is the use of the word “diagonalizable”: the Authors say multiple times that the dielectric tensor cannot be diagonalized. This is not entirely accurate, because matrices in the form of equation 1 can be diagonalized on a complex field in general, meaning that it is possible to find V and D such that $V^*D^*V^{-1}$ is equal to the original matrix. The special property here is that for the tensors considered in this paper V cannot be a real constant matrix (corresponding to a real rotation of the axes). Instead it has to be complex and frequency dependent. That implies that there is no system of coordinates (axes) that can be chosen so that both the real and the imaginary part of the matrix are diagonal. This point should be clarified in the main text, so it is clearer what it is meant by “non-diagonalizable”

We thank the reviewer very much for pointing out this issue. Indeed, the involved matrix can always be diagonalized in a complex basis, but the relevant transformations for our work are real-space geometrical rotations. The actual question we are raising is whether there is a reference frame for which the dielectric tensor can be made diagonal. It turns out that bGO does not generally allow such transformation when losses are considered (this is not true in the lossless limit). We have added a comment to the revised manuscript clarifying this point.

-Page 4: This sentence “In addition, the wavefronts are tilted with respect to the direction of energy flow, with no apparent mirror symmetry. This feature can also be clearly seen by examining the Fourier transform of the real-space profile (Fig. 1g), exhibiting a stronger intensity along one side of the hyperbola” is, I believe, not completely justified: it makes sense that the direction of the phase fronts is different from the energy flow, but figure 1g is not sufficient to justify this claim, since it is provided for a single frequency. A complete justification requires the calculation of the group velocity (which would be a very interesting addition to the study), which is done taking a differential in the frequency, so it requires calculation in at least two frequencies close to each other. Otherwise, only the phase velocity can be calculated. The asymmetry mentioned in this sentence arises from the fact that, considering a wave in a specific direction, the k vector has different attenuation (imaginary part) in the various quadrants identified by the hyperbola in the real part of k , as in figure 3. It would be beneficial and much more rigorous to plot real and imaginary part of k separately after solving Maxwells equation as opposed to the Fourier transform of the numerical full field simulations.

We very much appreciate the reviewer’s comment, and we reworded the respective text in the revised manuscript. They are correct in stressing the difference between the “direction of energy flow”, or energy velocity, and the phase velocity. Due to the modal dispersion these two quantities are not the same. Actually the energy flow is well described by the group velocity $\vec{v}_g = \nabla_{\vec{k}}\omega$, where $\omega = \omega(\vec{k})$ is a function of momentum \vec{k} . Therefore, we can retrieve the direction of group velocity and energy flow from our isofrequency lines, since \vec{v}_g is perpendicular to the tangential direction of the isofrequency dispersion lines, i.e., $\vec{v}_g \cdot d\vec{k} = 0$ where $d\vec{k}$ is an infinitesimally small displacement parallel to the tangential direction of the dispersion line.

In response to the reviewers comment, we have analyzed the group velocity in the analytical dispersion model (now included in the Suppl. Materials). Additionally we also analyzed the radial group velocity component along the simulated in-plane dispersion curves supporting the new experimental data in Fig. 2. These data (new Fig. 2j and new SI section 8) clearly reveal the asymmetric distribution of group velocities, thereby demonstrating the tilting of the direction of energy flow away from the axis of the hyperbola. We note that this phenomenon arises as a direct consequence of the continuous rotation of the major polarizability axes in bGO (and the resulting rotation of the hyperbola axis), and thus marks another representation of the shear effect.

We take the opportunity to stress that our results stem from solving rigorously Maxwell’s equations. Rather than solving them in the complex k space, for which it would not be easy to plot complex vector solutions, we do solve them for complex frequencies to assess the effect of loss and asymmetry on the polariton dispersion. The two approaches are equivalent and equally rigorous, but the complex frequency results allow an easier interpretation.

Referee #3 (Remarks to the Author):

In this work, Passler and co-authors describe theoretical and experimental work on identifying “hyperbolic shear polaritons” which they state can exist in low-symmetry crystals due to the presence of non-orthogonal IR-active phonon resonances. The work is novel and timely, and the manuscript appears to be well organized and well thought-out. I have provided some queries below that I had while reading the work, although generally I believe this work is well-suited for publication in Nature after revisions.

We thank the reviewer for their positive assessment of our work and the explicit recommendation of publication in Nature. In the following, we address the minor points raised by the reviewer.

1. In describing the complex dielectric tensor for the materials discussed, (“bGO” and “aQ”), the authors give a general form in Eq. 1. This is fine, however I am wondering what actually was used to compute these frequency-dependent terms. They presumably come from information related to the phonon eigenvectors and eigenenergies, which I imagine can be calculated using a first principles (ab initio) approach. Can the authors comment on how they arrived at their values for $\epsilon(\omega)$?

The dielectric tensor used for the simulations was derived from infrared ellipsometry measurements, see Ref. 7 of the manuscript. In this paper, the authors derived all phonon mode properties by modelling the dielectric response and suggested use of the eigendielectric polarizability model for the first time to correctly address transverse and longitudinal phonon modes in monoclinic crystals. The eigenvectors (phonon mode polarization direction) and eigenenergies (phonon mode frequencies) determined thereby from experiment then compared excellently with computational results obtained from first principles (density functional theory) calculations for monoclinic gallium oxide provided in the same reference.

2. There are several debates around phonon-polaritons in monolayers (for example see Nano Lett. 2019, 19, 4, 2653–2660 and Nano Lett. 2017, 17, 6, 3758–3763). Do any of the considerations presented there appear in the case of hyperbolic shear polaritons?

The reviewer raises an interesting point here. For bGO, we have not considered this question since it is not a natural 2D material. While exfoliation is possible, monolayer preparation of bGO has not been reported so far (to the best of our knowledge). It is interesting, though, to reassess the constantly growing material database of 2D materials for low-symmetry properties to then critically investigate this question. We added a comment and reference regarding phonon polaritons in monoclinic monolayers to the revised manuscript.

3. In describing this $\epsilon(\omega)$, the authors state that the off-diagonal terms $\epsilon_{xy}(\omega)$ and $\epsilon_{yx}(\omega)$ would be the same due to reciprocity. Later the authors state that $\text{Re}[\epsilon(\omega)]$ is Hermitian. What confuses me is then the final paragraph on page 7. At what point do things become “non-Hermitian”, as the authors suggest? Can the authors clarify what is meant by that, and whether or not the systems studied in this work are considered non-Hermitian? Although the paragraph introduces interesting ideas, I found it a bit confusing in terms of how it connected with the present work directly, so perhaps readers would appreciate a bit of a clearer connection.

We thank the reviewer for raising this question. $\text{Re}[\epsilon(\omega)]$ is Hermitian because it is symmetric and real-valued, and can therefore be diagonalized with real-valued eigenvalues. $\epsilon(\omega)$ of a real crystal,

however, even if diagonal, is non-Hermitian solely because of the losses represented by the imaginary components. For bGO, we face a special class of non-Hermitian tensor, where not only diagonal losses are present, but additionally off-diagonal loss terms result in new phenomena, as described in the paper. In other words, even though the real and imaginary permittivity tensors can be individually diagonalized by rotation, in bGO these rotations are different for both components, leading to situations where either the real or the imaginary off-diagonal components do not vanish. This brings new opportunities for topological phenomena that we have just started to explore.

Reviewer Reports on the First Revision:

Referee #1 (Remarks to the Author):

Summary of the key results

The work introduces shear polaritons as a new modality of these quasiparticles. It claims that new polaritonic features can emerge from bGO which are related to its monoclinic crystal structure. Continuous steering of propagation direction by frequency tuning, tilted wavefronts and asymmetric dielectrics are among the properties brought by this class of crystals, which are complementary to reported orthorhombic and hexagonal nanophotonic crystals. Therefore, it is claimed here that the discoveries presented can boost the polaritons physics field regarding low-symmetry materials, expanding the possibilities for devices developments.

I recommend publication in Nature in the current status of the manuscript.

Originality and significance:

The work is highly innovative and disruptive in the field. A new class of polariton is presented and comprehensively described. Moreover, the work consolidates an experimental procedure that can access in-plane momentum dispersion, opening unprecedented opportunities.

Data & methodology:

The work presents outstanding data regarding quality, clarity, presentation which makes the approach readily valid.

Appropriate use of statistics and treatment of uncertainties

The high Signal-to-noise ratio of the data didn't require sophisticated statistics analysis, therefore, I conclude the approach was appropriate.

Conclusions:

Conclusions are robust and coherently support all the claims of the work.

Suggested improvements:

Additional experiments requested during the review process indeed increased the robustness of the work making the work an experimental prime. Authors have carefully addressed the questions and included arguments increased the clarity of this scientific communication.

References: appropriate credit to previous work?

Yes

Clarity and context: lucidity of abstract/summary, appropriateness of abstract, introduction and conclusions

The text is lucid and very appropriate to communicate a disruptive advance.

Warmest regards and congratulations to the team for this exceptional work,

Raul O. Freitas
Brazilian Synchrotron Light Laboratory (LNLS)

Referee #2 (Remarks to the Author):

The Authors addressed all the technical points that I mentioned in the last review round. Specifically the calculation of the group velocity fully justifies the claims in the paper. I still have some doubts whether this paper is too specialized for the broader scientific community of Nature, but considering also the positive assesment of the other reviewers, i consider this point solved as well.

Referee #3:
No comments